# A simple cloud-filling approach for remote sensing water cover assessments

Connor Mullen [1], Gopal Penny [1], and Marc F. Müller [1]

[1]Department of Civil and Environmental Engineering and Earth Sciences at University of Notre Dame

**Correspondence:** Marc F. Müller (mmuller1@nd.edu)

**Abstract.** The empirical attribution of hydrologic change presents a unique data availability challenge in terms of establishing baseline prior conditions, as one cannot go back in time to retrospectively collect the necessary data. Although global remote sensing data can alleviate this challenge, most satellite missions are too recent to capture changes that happened long enough ago to provide sufficient observations for adequate statistical inference. In that context, the four decades of continuous global high resolution monitoring enabled by the Landsat missions are an unrivaled source of information. However, constructing time series of land cover observation across Landsat missions remains a significant challenge because cloud masking and inconsistent image quality complicate the automatized interpretation of optical imagery.

Focusing on the monitoring of lake water extent, we present an automatized gap-filling approach to infer the class (wet or dry) of pixels masked by clouds or sensing errors. The classification outcome of unmasked pixels is compiled across images taken on different dates to estimate the inundation frequency of each pixel, based on the assumption that different pixels are masked at different times. Inundation frequency is then used to infer the inundation status of masked pixels on individual images through supervised classification. Applied to a variety of global lakes with substantial long term or seasonal fluctuations, the approach successfully captured water extent variations obtained from *in situ* gauges (where applicable), or from other Landsat missions during overlapping time periods. Although sensitive to classification errors in the input imagery, the gap filling algorithm is straightforward to implement on Google's Earth Engine platform and stands as a scalable approach to reliably monitor, and ultimately attribute, historical changes in water bodies.

## 1 Introduction

The water extent of many lakes has changed substantially over the last few decades (Busker et al., 2019). Once imposing bodies of water have declined to a small fraction of their historical volume in many parts of the world, the Aral Sea standing out as an iconic example (Micklin, 2007). In more humid climates, shifts in the flow regimes of tributary streams has affected the seasonal variability of the corresponding lakes. For example, in the Mekong basin, changes in the seasonal flood pulse of the Tonle Sap threaten the lake's sensitive ecosystems and fishery (Kummu and Sarkkula, 2008) with direct repercussion

to the region's food security and unique biodiversity. These changes often emerge as a result of complex interplay of natural (e.g., changing temperatures and precipitations) and anthropogenic (damming and irrigation) factors (Haddeland et al., 2014). Proper attribution of their drivers is critical to inform policy, but is hampered by a dearth of monitoring data due to prevailing financial, institutional and legal barriers (Solander et al., 2016). In that context, a substantial body of recent research has focused on monitoring surface water extents using satellite data, with applications ranging from small reservoirs (Avisse et al., 2017; Gao et al., 2012; Zhao and Gao, 2018) to large water bodies (Mercier et al., 2002) at the regional (Müller et al., 2016), continental (Zou et al., 2018) and global scales (Busker et al., 2019; Pekel et al., 2016; Gao et al., 2012; Wang et al., 2018). By providing a consistent global space-time representation of the earth system, satellite imagery offers a unique ability to study and attribute change *ex post*, in situations where *in situ* observations are nonexistent, unavailable or disputed. Yet a sufficiently large sample of high quality remote sensing observations is necessary to attribute change with adequate statistical power (Müller and Levy, 2019). Hence, imagery used to monitor lake water extent to quantify long-term patterns needs to (i) cover a sufficiently long period of regular observations (e.g., several decades of monthly observations); and (ii) allow open water to be consistently distinguished from dry land at high spatial resolution in all weather conditions, including through clouds. These two requirements are challenging to satisfy simultaneously.

All-weather water detection can be achieved using active remote sensing at microwave frequencies. The process is unimpeded by clouds and does not rely on reflected sunlight. Synthetic aperture radars (SAR) in particular leverage the fact that areas of open, smooth water bodies exhibit lower back-scatter coefficients in the X, L or C bands (Bioresita et al., 2018). A number of recently launched SAR missions (e.g. COSMO-SkyMed, TerraSAR-X and Sentinel-1) allow for detection of water at resolutions and return times that are appropriate to capture local changes in water cover (Pérez Valentín and Müller, 2020). For instance, Sentinel 1 was launched in 2014 and has a 6-day revisit time and a spatial resolution of 20 m. However promising in their ability to monitor ongoing changes, these very recent sensors are unable to capture events that happened before their launch.

In contrast, satellites with optical sensors have been orbiting the earth for decades and remain a preferred source of information to monitor open water (see Huang et al., 2018a, for a recent review). A number of spectral indices have been proposed to detect water using multispectral imagery (see Zhou et al., 2017), including the Modified Normalized Difference Water Index (MNDWI, Xu, 2006) used in this study. These indices leverage the high contrast between land and water at specific frequencies of the electromagnetic spectrum, and a range of techniques have been developed to systematically classify pixels as "wet" or "dry" based on their spectral index (see Lu and Weng, 2007). A fundamental limitation of optical sensors, however, is their inability to capture land surface reflectance through clouds. A number of studies have addressed this impediment by leveraging the high (daily) return time of NASA's Moderate Resolution Imaging Spectroradiometer (MODIS) mission to build cloud-free lower frequency (e.g., bi-weekly) mosaics (Gao et al., 2012; Wang et al., 2018). MODIS has relatively short coverage period (1999 and 2002 to present for the Terra and Aqua satellites, respectively), but has been combined with space-borne radar altimeters to monitor lake water extents in earlier periods (up to 1992 using the Topex-Poseidon altimeter) by leveraging overlapping coverage periods to estimate water-level to inundation area relationships (Gao et al., 2012). However the limitations normally associated with radar altimetry (narrow swath, coarse cross-track spacing and large along-track path length (see Yale

et al., 1998)) have restricted this approach to lakes that are well covered by altimeter orbits (Gao et al., 2012). In addition, the
relatively coarse spatial resolution (250 m to 500 m for visible and near infrared bands) of MODIS limits its applicability for
smaller lakes. Unlike MODIS, the successive Landsat missions provide high resolution coverage of the earth surface since the
1970s. Landsat imagery has recently been used by Pekel et al. (2016) to generate consistent monthly 30-m resolution estimates
of global surface water cover (GSW) between the mid 1980s and 2015. However, Landsat image interpretation is complicated
by a set of well known challenges including clouds, cloud shadows, terrain shadows, and the Scan Line Corrector (SLC) fail-
ure on Landsat 7. These effects complicate the detection of surface water, causing approximately a third of the pixels in the
GSW dataset to be marked as 'no data' (see *Code and data availability*). Discarding these masked pixels when identifying
water-covered pixels will lead to a substantial underestimation of water cover (Zhao and Gao, 2018). This points to the need
for scalable and easily implementable post-processing approaches to infer the inundation status of masked pixels.

We address this problem by predicting the binary class (e.g., 'wet' or 'dry') of masked ('no data') pixels, based on the
observed class of comparable unmasked pixels. Two broad sets of such gap-filling approaches have been proposed in the
literature. The first set of approaches are based on topographic consistency: a pixel will not be 'dry' if it lies at an elevation
that is lower than the highest (unmasked) 'inundated' pixel within the same water body (Khandelwal et al., 2017; Avisse et al.,
2017). An important limitation to these approaches is their reliance on either a digital elevation model (Khandelwal et al., 2017;
Avisse et al., 2017) or a radar altimeter (Van Den Hoek et al., 2019). However, digital elevation models can have a low level
of accuracy in the vertical direction (with standard deviation on the order of meters (Avisse et al., 2017)) and may not capture
the topography of regions that were flooded during the satellite overpass, whereas radar altimeters are limited with the spatial
coverage limitations that we previously discussed (Yale et al., 1998). In contrast, the second set of studies does not rely on
ancillary information but uses the historical inundation frequency (IF) of a masked pixel (estimated using observations taken
at times when the pixel was unmasked) to predict its current inundation status. Zou et al. (2018) use a fixed IF threshold of
0.75 (i.e. pixels that are inundated on 75% or more of the unmasked images) to identify permanent water bodies. Zhao and Gao
(2018) apply a heuristic on the histogram of the IF of unmasked inundated pixels: masked pixels with an IF value larger than
the IF corresponding to an arbitrary (i.e. 0.17) fraction of the mean histogram value are classified as 'inundated'. Schwatke
et al. (2019) use an IF-image as a proxy for a digital elevation model and estimates an area-IF curve for each lake as a proxy
for its area-elevation curve. An iterative algorithm is then used to estimate the maximum IF value of masked inundated pixels,
so as to maintain topographic consistency within the lake.

Here, we present a new method for cloud-filling remotely sensed time series of surface water. In particular, we use a super-
vised classification technique to infer a statistical relationship between the IF value and the inundation status of the unmasked
pixels, which we then use to predict the inundation status of the masked pixels of the same image. Unlike Zou et al. (2018)
and Zhao and Gao (2018), the proposed approach does not rely on arbitrary heuristics but uses information from *all* unmasked
pixels (both inundated and dry) to infer the status of masked pixels. Unlike Schwatke et al. (2019), the approach is exclusively
based on pixel-level statistical relationships and does not rely on aggregate-level constraints such as maintaining topographic
consistency within the lake. This feature allows it to use a standard machine learning technique (random forest) and leverage
the massive parallelization capability of Google Earth Engine, thus benefiting from the scalability and portability associated

with that platform. The approach is independent from cloud and water classification approaches that are used to construct the
ternary images (i.e., images comprised of 'wet', 'dry', and 'no data' values) used as input, and our results demonstrate that
gap-filling performance is generally robust to unbiased classification errors.

The proposed gap-filling algorithm is described in Section 2.1, along with its four underlying assumptions. These assumptions then structure the validation of the approach. We first assess its sensitivity to deviations from each assumption through the numerical experiments described in Section 2.2, with results presented in Section 3.1. We then evaluate the propensity for such deviations to happen in practice by applying the approach to monitor the extent of 9 global lakes using Landsat 5, 7, and 8 imagery. The selected lakes represent a variety of sizes and climatic and topographic characteristics and were selected based on the availability on *in situ* data (Section 3.2) or documented historic water extent variations (Section 3.3). Section 4 discusses the results and offers concluding thoughts on the specific contribution of the proposed method with regard to other existing gap-filling algorithms. We also provide a JavaScript function that can be readily integrated into any Google Earth Engine script (see *Code and data availability*)

## 2 Methods

### 2.1 Gap filling algorithm

The algorithm addresses the challenge of converting a time-series of ternary images ('wet', 'dry', and 'masked' categories, Figure 1B) into an equivalent time series of binary images ('wet' and 'dry', Figure 1D). To do so, it uses a readily available supervised classification method (random forest, Pelletier et al., 2016) to infer the category ('wet' vs. 'dry') of masked pixels based on their inundation history. For this purpose, space-time information about historical water extents are compiled into a single inundation frequency (IF, Figure 1C) image representing the historic probability of each pixel location $i$ being categorized as 'wet' across the time series of images:

$$IF_i = \frac{N_i^{(wet)}}{N_i^{(wet)} + N_i^{(dry)}} = \frac{N_i^{(wet)}}{N - N_i^{(masked)}} \tag{1}$$

where $N_i^{(wet)}$, $N_i^{(dry)}$ and $N_i^{(masked)}$ are (respectively) the number of times pixel $i$ appears as wet, dry or masked over the $N$ considered monthly images. For each image, the supervised classification algorithm then proceeds to estimate a statistical relationship between the inundation status of unmasked pixels and their IF value. This relationship is then used to infer the status of all pixels of the image based on their own IF value. Classification noise that emerges from the uncertainty of the estimated statistical relationship is then dampened through morphological filtering (Schowengerdt, 2006). Note that the supervised classification algorithm is run independently on each individual image using a different set of unmasked classified pixels as training (depending on the associated cloud mask), but using the same IF image as predictor. An implementation example using monthly ternary images ('wet','dry','masked') from Pekel et al. (2016) is provided under *Code and data availability* below.

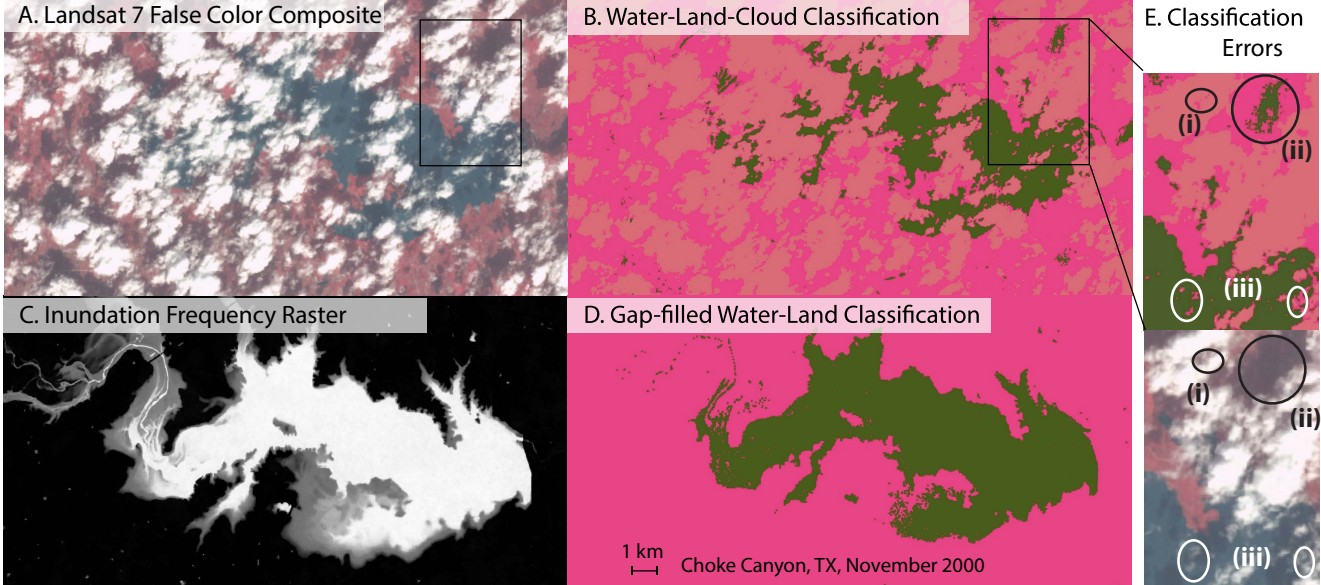

**Figure 1.** Illustration of the gap filling algorithm. **A.** Original Landsat 7 false color composite image for Choke Canyon, TX, November 2000. **B.** Input classified ternary image from (Pekel et al., 2016) with the wet, dry and masked classes represented in green, dark pink and light pink, respectively. **C.** Inundation Frequency image constructed using the 430 monthly ternary images from the (Pekel et al., 2016) between March 1984 and December 2019. The IF value is displayed on a linear scale of grays with values of 0 and 1 respectively represented as black and white. **D.** Output binary image for November 2000, with wet and dry pixels represented in green and dark pink, respectively. **E.** Examples of classification errors: (i) Light clouds over land mistakenly classified as clouds; (ii) land in cloud shadows mistakenly classified as water; (iii) light cloud over water mistakenly classified as land.

A fundamental assumption of the approach is that pixels with a higher IF-value are lower topographically, and therefore
more likely to be inundated on any given image. More specifically, if unmasked pixels associated with a certain $IF$ value are inundated in a given image, it is very likely that pixels with an equal or higher IF value (i.e. pixels of equal or lower elevation) are also inundated. This assumption holds if four important conditions are satisfied:

– **Classification accuracy:** The ternary input image must be accurate in that the classification technique accurately distinguishes water, land and no data in the original multispectral imagery. An overly eager cloud detector would mask
too many pixels and decrease the precision of the supervised classification in the gap-filling process. An overly cautious cloud detector (or a faulty water detector) would lead to misclassification of water (or clouds) as land, and vice-versa. This then affects gap-filling by introducing errors in both the IF raster and the classification of unmask pixels in individual images used to train the supervised classifier.

– **Independence:** The propensity of a pixel to be masked in any given image must be independent of its inundation
status. If this assumption does not hold, the inundation status of a pixel determines its cloud coverage. Under these

conditions, the relationship between its IF and inundation status estimated in cloudless conditions will not reliably predict its status in cloudy conditions. This situation may arise, for instance, from fog being produced by the microclimatic conditions associated with open water (Koračin et al., 2014), or from spatially persistent classification errors associated with topographic shading (Huang et al., 2018b).

- **Stationarity:** The statistical relationship between the IF value and the inundation status of pixels must not change over time. A threat to the stationarity assumption might emerge, for instance if erosion or sedimentation processes substantially alter the near-shore bathymetry of the lake.

- **Homogeneity:** The statistical relationship between the IF value of pixels and their inundation status must be homogeneous in space. This assumption is necessary for the IF-inundation status relationship estimated for unmasked pixels to be transferred and applied to mask pixels. This could be violated in situations when the lake bathymetry contains multiple depressions and the lake separates into multiple water bodies as water levels fall.

## 2.2 Validation

A direct validation of the approach would require a sample of *in situ* observations of lake extents that (i) is representative of the variety of water bodies that the method applies to, and (ii) matches the monthly frequency and multi-decadal observation period that are targeted by the analysis. The few openly available datasets that span such long observation periods typically focus on small to medium sized regulated reservoirs within the US) and/or feature lake elevation time series with no reliable elevation-area relationships to estimate lake extents. To address this data availability challenge we use a two-step validation approach focusing on the four main error sources identified in the previous section. In the first step, we investigate the sensitivity of the gap filling algorithm to each error source using numerical experiments (Section 2.2.1). In the second step we illustrate the application of the approach to monitor the water extent of real lakes and discuss the propensity of each error source to emerge in real life. The approach is implemented on a sample of 9 particular lakes that span a variety of sizes, geographic locations and levels of data availability (Section 2.2.2).

### 2.2.1 Numerical Experiments

We use numerical experiments to evaluate the sensitivity of the gap-filling approach to deviations from its four fundamental assumptions. The experiments use 430 monthly ternary classification images (wet, dry, masked) obtained from Pekel et al. (2016) for Choke Canyon Reservoir (TX) between March 1984 and December 2019. Note that the experiment hinges on the controlled addition of random classification errors, and is not materially affected by the specific location chosen as a baseline. The numerical experiment then proceeds as follows:

1. A fraction $F_1$ of unmasked pixels in each image is randomly selected and masked.

2. A fraction $F_2$ of the remaining unmasked pixels in each image is then (independently) randomly selected and flipped, i.e. recast as 'wet' if they are 'dry' and vice versa.

3. The gap-filling algorithm is then carried out using the appropriate combinations of images from steps 1 and 2 (see below) to construct the IF raster and the training dataset.

4. The predicted inundation status ('wet' or 'dry') of the pixels masked in step 1 are compared to their original status. The proportion of masked pixels that are misclassified in the gap-filling process is recorded as gap-filling error. We finally compute the mean gap-filling error across images and its 95% empirical confidence interval.

We carried out the following experiments to simulate deviations from each of the four assumptions (see *Code and data availability* below):

– **Classification accuracy:** We simulate the effects of (i) over-detection of cloud and (ii) under-detection of clouds or misclassification of land as water (and vice versa) by respectively (i) varying the fraction $F_1$ of unmask pixels in step 1 and (ii) varying the fraction $F_2$ of 'flipped' pixels in step 2. We simulate the combined effect of both types of errors by considering combinations of $F_1$ and $F_2$.

– **Independence:** We evaluate the effect of a correlation between the IF value of the pixels and their inundation status by comparing the outcome of two experiments. In the first (baseline) experiment, the pixels flipped in step 2 are independently drawn for each image. In the second (alternative) experiment, the pixels flipped in step 2 are drawn once and do not vary across images. Because the flipped pixels are persistently wrongly classified in the alternative experiment, we expect a persistent bias to emerge in the relationship between IF and inundation status estimated by the supervised classifier. This, in turn, will lead to a larger gap-filling error compared to the baseline experiment. We measure the effect of a non-independent inundation status as the difference between the gap-filling errors associated with the alternative and baseline experiments.

– **Stationarity:** We simulate the effect of an IF-inundation status relationship that evolves over time by *only* introducing errors in the images used to construct the IF-raster. We introduce persistent errors in step 2 by flipping the same pixels on all images, which we then use to construct the IF raster. However, we use the outcome of step 1 (the *unflipped* images) as training data when carrying out the supervised classification in step 3. This represents the situation where an outdated (here, noisy) IF is being used to classify contemporaneous observations. The larger the percentage of pixels flipped, the 'noisier' the IF and thus the less representative it is of the actual IF of the training images. Under these conditions, the simulated gap-filling errors represent the effect of violating the stationarity assumption.

– **Homogeneity:** We simulate the effect of a spatially heterogeneous IF-inundation status relationship by introducing a persistent error in the training data but *not* in the images used to construct the IF raster. Under these conditions, the relationship between the IF value and the inundation status that prevails for the unflipped pixels will be inverted for the flipped pixels. This portrays a situation where an arbitrary subset of pixels with a given IF value will tend to be wet whenever the remaining pixels with the same IF value are dry, as can emerge for example in a wetlandscape where water bodies are governed by the same hydrologic drivers when connected and different drivers when disconnected

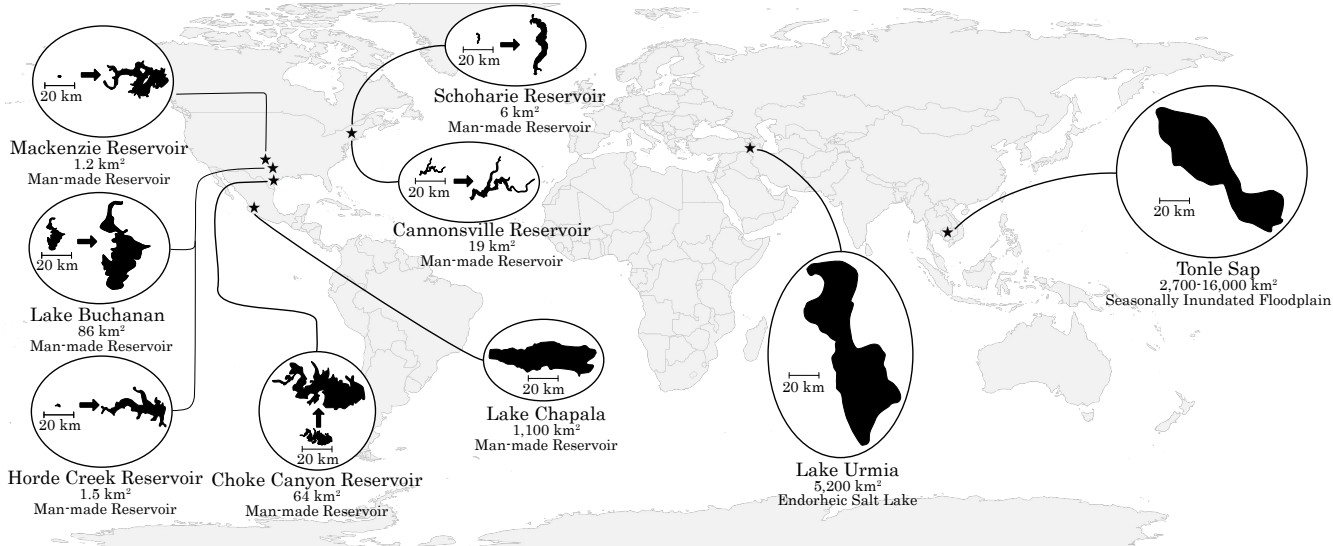

**Figure 2.** Location and characteristics of the considered water bodies.

(e.g., drainage vs. seepage). In that context, the fraction $F_2$ of pixels flipped represent the degree of heterogeneity of the
landscape (i.e. 50% means that half the pixels are governed by an inverted IF-flooding status relationship)

### 2.2.2  Application to real lakes

We focus on 9 particular lakes – 6 gauged lakes in the US, and 3 ungauged lakes outside the US (Figure 2) – to illustrate the
practical application of the gap-filling algorithm (Section 3.2), and discuss the validity of its four underlying assumptions in
operational situations (Section 4). The 6 US lakes have between 17 and 47 years of daily water level observations, available
from the United States Geological Survey and the Texas Water Development Board. The four water bodies in Texas are
emblematic of changing seasonal to inter-annual lake conditions that prevail in intensively managed small lakes and reservoirs
in semi-arid areas. The two reservoirs in upstate New York represent the complex topography and strongly seasonal climate
and land-cover (including snow and ice) that prevail in high latitude mountainous regions and complicate cloud and water
detection. For each lake, monthly water extents were determined based on daily water levels using the provided elevation-
area-capacity tables and corrected for additive bias (see Supplementary Information, SI). The three lakes outside the US have
documented seasonal and inter-annual changes in their water extents that are of major regional significance: Lakes Tonle Sap
(Cambodia), Urmia (Iran) and Chapala (Mexico). No long-term *in situ* observations were available for validation. However, we
compared estimates from the Landsat 7 to estimates from Landsat 5 and 8 during respective overlapping periods. This process
provides reasonable estimates of lake extent prediction errors, assuming that sources of errors across Landsat missions are
close to independent (different sensors on different space platforms taking images at different times, see Table 1).

Input to the gap-filling algorithm can be provided by any cloud- and water-detection method that is able to generate the required input ternary images. Here, we demonstrate its application using two particular techniques that are widely used in practice and straightforward to implement on Google Earth Engine, noting that more elaborate approaches to detect both clouds (Foga et al., 2017) and water (Rokni et al., 2014; Lu and Weng, 2007) on Landsat imagery are available. A rudimentary cloud-scoring algorithm available on Google Earth Engine (`ee.Algorithms.Landsat.simpleCloudScore()`) is used to detect and mask clouds based on Top of Atmosphere Landsat reflectance images. Pixels indicated as faulty (e.g., due to the Landsat 7 Scan Line Corrector failure) are also masked out. The weekly to bi-weekly Landsat images are then aggregated at the monthly time scale through maximum value compositing using the Normalized Difference Vegetation Index (NDVI) (Chen et al., 2003). This last step is based on the presumption that clouds have a low NDVI value. Cloud-free pixels of each monthly image were then classified as wet or dry based on their modified normalized difference water index (MNDWI) value (Xu, 2006), that is the normalized difference between the green and mid-infrared bands of the relevant Landsat sensor (see Table 1 for corresponding bands in the considered imagery). The MNDWI enhances water/land contrasts by leveraging the ability of open water (compared to dry land) to preferentially absorb and reflect in the mid-infrared and green regions of the electromagnetic spectrum, respectively. A clustering algorithm is applied to each image to identify the MNDWI threshold that partitions its pixels into two sets, so as to minimize the MNDWI variance within each set. Because it can dynamically separate dry and wet pixels in cloud-free images, unsupervised classification stands as a promising (and somewhat less arbitrary) alternative to the manual determination of classification thresholds implemented in past studies (e.g., Müller et al. (2016) among others). However, by minimizing within-cluster variance, k-means tends to favor clusters of comparable sizes (Jain, 2010), which is problematic for cloudy images with preferential cloud covers on either land or water. As an extreme example, if all unmasked pixels are covered by water, a two-cluster k-means classification will not be able to distinguish water from land. We address this issue by computing the median value from the set of MNDWI thresholds obtained from the classification of individual images. This single median MNDWI threshold is then used to (re)classify all unmasked pixels from all monthly images. Assuming the unsupervised classification can distinguish water from dry land on *most* images, the median threshold will allow for the identification of all unmasked pixels from the above extreme example as "wet". A time series of lake area is finally generated by counting, on each monthly classified image, the number of inundated pixels within a predetermined polygon encompassing the maximum historical extent of the lake. Outlier predictions associated with detection errors (see Section 4) are automatically identified and removed using the approach described in (Chen and Liu, 1993).

## 3   Results

### 3.1   Numerical Validation

Results of the numerical experiments are presented in Figure 3. Panel A displays gap-filling errors for various combinations of $F_1$ (pixels masked) and $F_2$ (pixels flipped). The former ($F_1$) represents the effect of the supervised classifier being provided with 'too little' information in the sense that the cloud detector overestimates cloud coverage. Results in Figure 3A suggest that this has a modest effect on gap-filling errors as long as the remaining (unmasked) pixels are correctly classified as water

| Satellite | Spectral Bands | Resolution | Return Time | Coverage |
|---|---|---|---|---|
| Landsat 5 | B2-Green (0.52-0.60 $\mu m$)<br>B5-MIR1 (1.55-1.75 $\mu m$) | 30 m | 16 days | 1984 - 2013 |
| Landsat 7 | B2-Green (0.52-0.60 $\mu m$)<br>B5-MIR1 (1.55-1.75 $\mu m$) | 30 m | 16 days | 1999 - Present |
| Landsat 8 | B2-Green (0.53-0.59 $\mu m$)<br>B6-MIR1 (1.57-1.65 $\mu m$) | 30 m | 16 days | 2013 - Present |

**Table 1.** Properties of Landsat data sources

or land. Introducing even modest levels of classification errors in the unmasked pixels (e.g, $F_2 = 5 - 10\%$ of unmasked pixels
are flipped) can cause the gap-filling error to blow up for high levels of $F_1$. In other words, for sufficiently high cloud cover
or small lake size, the accuracy of the approach becomes highly sensitive to classification errors, which occurs in the example
when more than 75% of the lake is masked. Given that the lake in the synthetic analysis is $\sim 1km^2$, precautions should be
taken when lakes are covered by excessive clouds or lakes are sufficiently small such that unmasked pixels cover less than 25
ha (or roughly $17 \times 17$ Landsat pixels).

These classification errors are further investigated in Figure 3B. Of note is that gap filling errors arising from water-land
classification errors (Figure 3B) are generally larger than those arising from an overestimation of cloud cover (Figure 3A).
This suggests that the gap filling approach works best combined with an overly eager cloud detection algorithm that tends
to overestimate (rather than underestimate) cloud cover. Importantly, Figure 3B also suggests that the gap-filling approach is
generally robust to faulty water-land classification in input images. Introducing classification errors into up to $F_2 = 30\%$ of
260 unmasked pixels of each image causes gap-filling errors in less than 10% of the control pixels. For context, a value of $F_2 = 50\%$
would represent the situation where wet and dry pixels are perfectly randomly distributed throughout the image (white noise).
An $F_2$ value larger than 50% reintroduces some signal; in particular $F_2 = 100\%$ has the same information as $F_2 = 0$, but with
all 'wet' and 'dry' pixels being swapped. The numerical experiment also allows to assess the pathway through which input
classification errors affect gap filling performances. Specifically, the supervised classification is affected by (i) errors in the IF
raster used as a predictor of inundation status for all images, and by (ii) errors in the individual images used by the classifying
as training. We investigate the relative importance of these two pathways by using the 'flipped' images from step 2 (see Section
2.2) to *either* construct the IF raster *or* serve as training data for the classifier; unflipped images from step 1 are then used to
fulfill the other task. Results in Figure 3B suggests that the gap filling algorithm is more sensitive to classification errors in its
training data (blue) than to errors in its IF raster (green).

Results in Figure 3C indicate the sensitivity of the gap filling approach to deviations from each of its four underlying
assumptions. The approach is most sensitive to errors in the detection of water and land in the input ternary imagery, although
diversions from all four assumptions have a generally modest effect on gap-filling errors. As in Figure 3B, gap-filling errors
remain below 10% for up to 30% of pixels flipped (note that red symbols in Fig 3B and 3C have an identical meaning).

For higher levels of deviations (>30% of pixels flipped), deviations from the independence (blue) and homogeneity (green) assumptions have comparable effects, which is both lower than that of classification errors (red) and higher than that of non-stationarities (purple). Note that the experiments used to evaluate stationarity and homogeneity assumptions are similar to the experiments to distinguish the IF-errors from training errors on Figure 3B, with the important distinction that the errors introduced to evaluate the assumptions are persistent in space (i.e. they are not independently drawn for each input image). The negative values in the gap filling errors obtained for the independence experiment (blue) arise from image-by-image subtraction of classification errors that is included in the experiment (see Section 2.2): for particular images, the gap filling error obtained from independently drawn classification errors is ostensibly larger than that obtained from persistent classification errors.

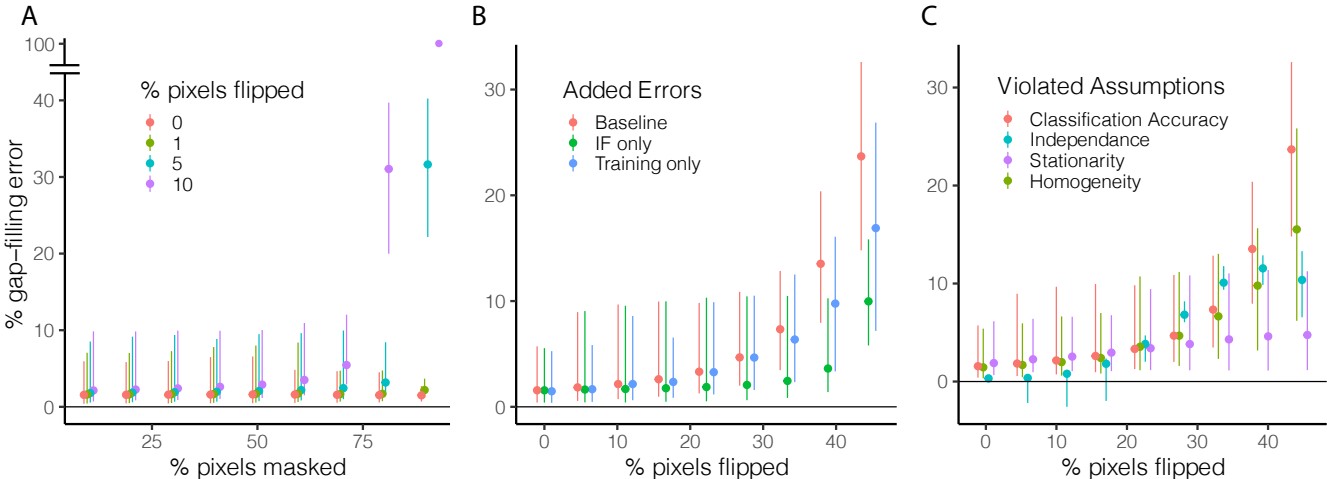

**Figure 3.** Results of the Numerical Experiments. **A.** Gap filling errors resulting from various combinations of independent random errors in cloud (% pixels masked) and water (% pixels flipped) detection. **B.** Origin of the gap filling errors associated with faulty land/water detection. Images with introduced errors are alternatively used to construct the IF-raster (green) or the training dataset (blue) or both inputs (red) of the supervised classifier used to estimate the status of masked pixels. **C.** Effect of deviations from the four fundamental assumptions obtained from the four numerical experiments described in Section 2.2. The % gap filling errors in panels B and C were evaluated by masking 5% of unmasked pixels in each image. These pixels were than used as validation data (step 1 in Section 2.2). Validation pixels were randomly and independently sampled for each image.

## 3.2  Application to real lakes

Applications to US lakes with available *in situ* lake level observations are presented in Figures 4 (Horde TX, Choke TX and Cannonsville NY) and S1 (Buchanan TX, Mackenzie TX and Schohaire NY), where the gap-filling algorithm was combined with commonly implemented approaches to detect clouds and water on multispectral Landsat imagery. Lake area outputs from 7 (brown) generally fit bias-corrected water extent estimates (black) based on lake level observations, suggesting that the remote sensing approach was able to capture the strong temporal change in water extent of these intensively managed reservoirs. Of

note is that the outlier predictions, which were removed *without* user input (following (Chen and Liu, 1993), and displayed as crosses on Figures 4 and S2) predominantly concern lakes in upstate New York and are clustered in the winter season (shaded on Figures 4 and S2). This points to known challenges in detecting open water in a landscape where land (and sometimes water) are covered by snow (e.g., Acharya et al., 2018). These challenges and their implications for the gap-filling algorithm are further discussed in Section 4. After removing winter classification results, lake extents estimated from Landsat 7 were strongly correlated to *in situ* observations for all lakes (Figures 4 and S1).

Application to Lakes Tonle Sap, Urmia and Chapala, where no *in situ* observations are available, shows a high level of agreement across Landsat missions during overlapping periods (Fig. 5). The analysis suggests that recent fluctuations in the amplitude of the seasonal inundation cycles of the Tonle Sap, which are critical to maintain its function as a regional biodiversity and food security hotspot, are decreasing. This is consistent with recent modeling simulations that predict decreased seasonal variations owing to flow regime alterations in the Mekong tributary region (Yu et al., 2019; Kummu and Sarkkula, 2008). The dramatic desiccation of Lake Urmia, once among the world's largest freshwater lakes, is also clearly visible in our analysis. Lake extent has declined steadily since the late 1990s to reach a low point in August 2014, which is consistent with existing estimates (AghaKouchak et al., 2015). Similarly, large water fluctuations in Lake Chapala, a strategic and historically over-exploited reservoir in Central Mexico (Wester, 2008; Godinez-Madrigal et al., 2019) in the 1990s and early 2000s can be seen in our analysis, along with the effects of the dramatic (albeit controversial (Godinez-Madrigal et al., 2019)) remediation policies that were implemented thereafter to restore lake levels (Wester, 2008).

## 4  Discussion

Results from the numerical experiments suggest that the performance of the gap-filling algorithm is generally robust to deviations from its four underlying assumptions. However, the analysis also showed that performance can be strongly impacted if these deviations are substantial enough. Therefore, the propensity of these four deviations to emerge in practice is an important question to consider when validating the proposed approach.

- **Classification Accuracy:** Despite its widespread use, the identification of clouds and water based on spectral indices entails inherent limitations. For example, challenges in distinguishing open water pixels from cloud or topographic shadows, or from snow-covered land, based on their MNDWI value have been reported in the literature (see e.g., Zhang et al., 2015; Huang et al., 2018b) and encountered in our analysis (Figure 1E). However, the lack of direct *in situ* observations of lake extents and the highly local nature of the error source (e.g., topography, snow cover) makes it challenging to estimate their general prevalence. Instead, we find it helpful to characterize classification errors as having two distinct and alternative effects. On the one hand, misclassification of either land or water as clouds, for instance due to an overly eager cloud detector, will decrease the amount of input information (*too little* information). On the other hand, misclassification of water (or land) as land (or water) will introduce an error into the input information (*wrong* information). This situation can emerge from an overly cautious cloud detector, where undetected clouds are then arbitrarily classified as either water or land. Results from the numerical experiments suggest that *wrong* input information

has a much larger effect on the gap-filling performance than *too little* input information (compare red symbols in Figures 3A and B). This insight is corroborated by comparing two sets of lakes from the case studies. The approach performed well for the two small lakes in Texas (Horde Creek and Mackenzie reservoir, $\sim 1km^2$ each), where the semi-arid climate and the flat topography are not prone to water classification error, but their small size limits the number of input pixels (*too little* information). In contrast, the two lakes in upstate New York (Schoharie and Cannonsville reservoirs) have more input pixels but the cold climate and mountainous terrain introduce errors in the unsupervised classification of water and land (*wrong* information). There, the gap filling algorithm performed markedly worse, particularly in winter when snow and ice are prevalent. These results illustrate a key limitation of the approach, that gap-filling accuracy is constrained by the accuracy of the input ternary imagery. They also suggest that the approach is more compatible with an overly eager cloud detector: by overestimating cloud cover the input imagery will err in favor of providing too little (rather than wrong) information, which has a smaller effect on the accuracy of the gap filling algorithm. The benefits of an over-eager cloud detection algorithm will be limited when unmasked pixels cover a sufficiently small area (roughly 20-30 ha), at which point accuracy becomes highly sensitive to *wrong* information.

– **Independence:** A threat to the independence requirement may emerge if the inundation status of a pixel determines its cloud coverage. For instance, fog can be produced by the micro-climatic conditions associated with open water (Koračin et al., 2014). We test whether threats to the independence assumptions emerged in our case studies by comparing the inundation frequency of pixels during cloudless days, with their inundation frequency estimated for *all* days. The former corresponds to the IF value from Equation (1). The latter was determined computing the estimated IF values of pixels *after* gap-filling, which includes cloudy days. We sampled 4000 pixels with IF values between (and excluding) 0 and 1 for both images (before and after gap-filling). We then ranked the pixels according to their IF value for each image. The independence assumption implies that the pixel rank is not affected by its cloud coverage status: a pixel with a higher inundation frequency than another for a subset of observations that had cloudless conditions should also have a higher inundation frequency if the full sample of observations (cloudless and cloudy) is considered. Results, shown in Fig. 6 (Top), suggest that the ranking of inundation frequency does not depend on cloud coverage. In other words, the independence assumption does not appear to be threatened in the considered lake. Note that non-random cloud coverage will only affect classification output if it concerns pixels near shores (i.e. where $0 < IF < 1$). This excludes permanently inundated pixels, which are predominantly affected by fog-over-water (Koračin et al., 2014).

– **Stationarity:** We used a split sample approach to determine whether the relationship between IF and the inundation status of pixels remains constant over time. Two IF images were constructed using the first (1998-2009) and second (2010-2020) half of the available Landsat 7 images. The inundation frequencies given by the first and second IF images were then collected for a random sample of 5000 pixels with $IF \in ]0, 1[$ on both images. The sampled pixels were then ranked according to their IF value for each image. The stationary assumption implies that the rank of the pixels does not vary between the two observation periods: If bathymetry did not change, a pixel that is more often inundated than another pixel during the 1998-2009 period should still be more often inundated during the 2010-2020 period. Results

on Fig. 6 (Bottom) suggest that the effect of bathymetric change on the classification outcome is negligible. Note that classification outcomes are only affected by bathymetric changes that concern those pixels that lie within the range of variability of water extent. This excludes pixels that are permanently covered (IF=1), where bathymetry may be most affected by sedimentation processes.

– **Homogeneity:** The homogeneity assumption implies that the relationship between the historical inundation frequency
of a pixel and their current inundation status does not vary in space. In other words, pixels that are historically more often inundated will be more likely inundated on any given day. This assumption clearly holds for the non-disjoint bodies of water that are considered in this study, but may not apply to bodies of water that fragment upon drainage (Figure 7). There, the gap filling algorithm should be applied independently for each homogeneous region. The need to identify homogeneous regions *a priori* in fragmenting lakes and more complex wetlandscapes is an important limitation of the
approach.

## 5   Conclusion

We propose a gap-filling approach that uses a standard supervised classification algorithm to predict the binary status (wet-dry) of masked pixels based on the historic frequency of their status. We validate the approach by (i) using numerical simulation to assess its sensitivity to deviations from its four fundamental assumptions and (ii) applying it to 9 global lakes representing a
variety of sizes, climates, topographies and levels of *in situ* data availability. Applying the approach to real lakes also allows us to evaluate the propensity for fundamental assumptions of the approach to hold in practical situations. Both analyses suggest that the approach is robust to substantial deviations from its underlying assumptions, several of which are likely to hold in most practical settings. However, the analyses also outlined two important limitations of the approach. First, the approach is sensitive to classification errors in the input imagery, particularly in small lakes. Misclassification of the output binary classes
(here wet/dry) have a stronger impact on performance than misidentification of masked pixels (here clouds) and the effect is exacerbated when unmasked lakes pixels fall below 25 ha (roughly $17 \times 17$ Landsat pixels). This further implies that the approach might not perform well in locations where circumstances (topographic shading, cloud shading, snow/ice, etc) makes it difficult to reliably distinguish water from clouds and land using multispectral imagery. In contrast, the method appears generally robust to situations where a limited number of input classified pixels are available for training (e.g., small lakes or
high cloud coverage). These two observations imply that the approach is preferably combined with a cloud detector that tends to overestimate cloud coverage. Second, the approach requires the *a priori* identification of homogeneous regions, where the relationship between the inundation frequency and inundation status of pixels is unique. This requirement limits the scalability of the approach in complex wetlandscapes, where the relationship might vary through space.

Despite these limitations, the approach stands as a promising approach (it can be readily implemented in Google Earth
Engine, see *Code and data availability*) to monitor the water extent of lakes and reservoirs at scale, particularly when combined with recent global datasets of ternary (wet/dry/masked) water cover (Pekel et al., 2016; Donchyts et al., 2016). More generally,

the algorithm can be used to infer the status of *any* masked binary imagery (not only water cover) that satisfies its four fundamental assumptions.

*Data availability.* Lake level datasets for validation are publicly made available by the United States Geological Survey (https://waterdata.
usgs.gov/nwis) and the Texas Water Development Board (https://www.waterdatafortexas.org/reservoirs/statewide)

*Code and data availability.* Basic implementation of the gap filling algorithm applied on ternary images from (Pekel et al., 2016): https: //code.earthengine.google.com/767ad2fe5931857550056e41213a4dcb. Gap filling algorithm combined with MNDWI-based classification of Landsat 7 images:https://code.earthengine.google.com/49efc5e51b9257da9a72d45c8ce486be. Numerical experiments used to test the four underlying assumptions: https://code.earthengine.google.com/1d7e23f5d5594ff9574fa73dd651b52e. Analysis of the percentage of masked
pixels in the (Pekel et al., 2016) dataset: https://code.earthengine.google.com/b41fdccbe6267d6a7e4c40deae8e9bf5

*Author contributions.* C.M. and M.F.M designed the research, C.M. and G.P. conducted the analysis, C.M. and M.F.M wrote the paper

*Competing interests.* The authors declare that they have no conflict of interest

*Acknowledgements.* We acknowledge financial support from the US National Science Foundation (NSF) under grant ICER 1824951.

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

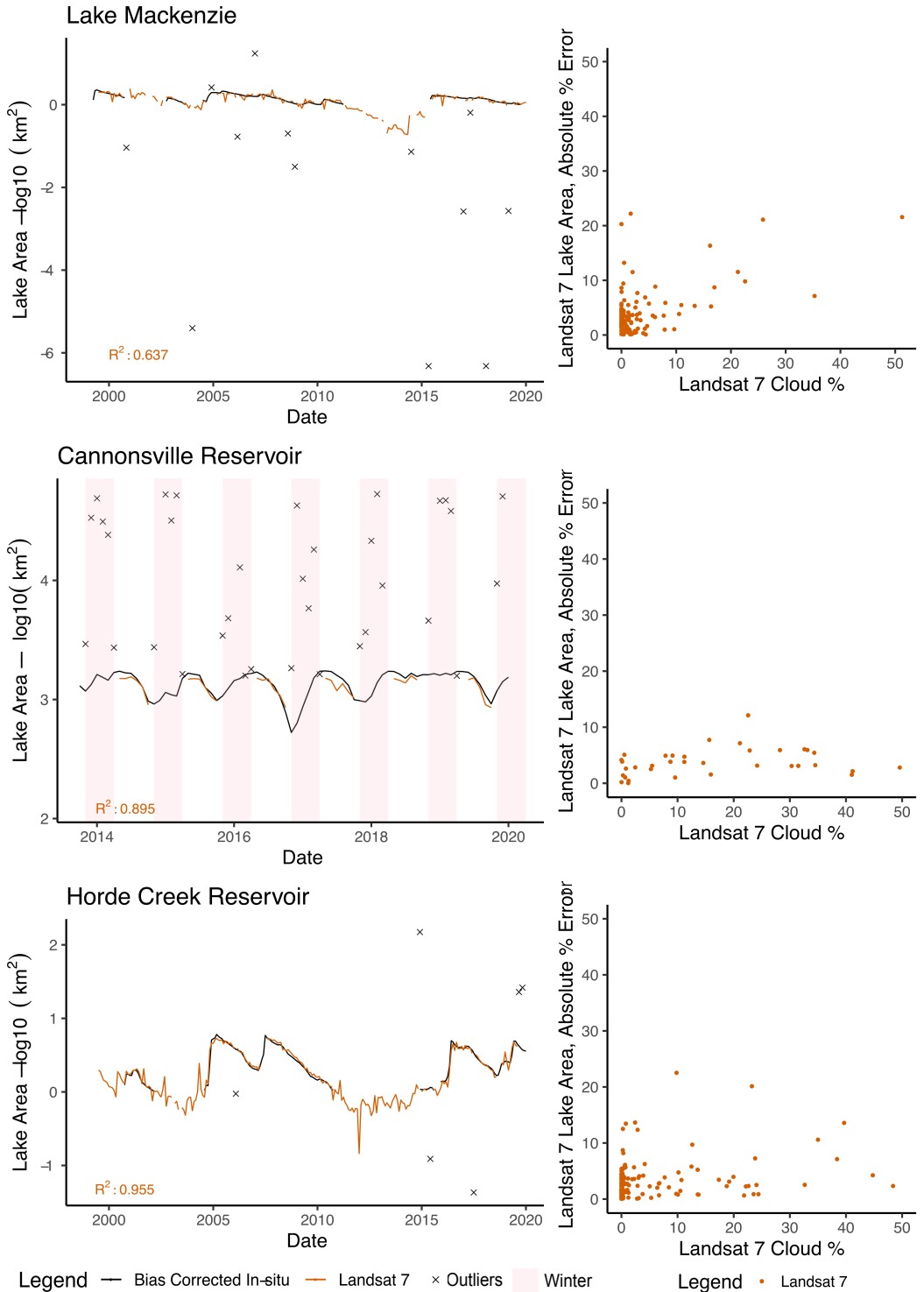

**Figure 4.** Application to lakes with *in situ* observation data *Left*: Time series representation water extent from *in situ* observation (black) and Landsat 7 (brown). Automatically removed outliers (crosses) are also displayed for indicative purposes. Winter months (December to February) are shaded out for Cannonsville NY. *Right* Scatter-plot of absolute percentage errors on Landsat 7 water extent estimates (compared to in situ observations) against the proportion of the lake's maximum footprint that was covered by clouds.

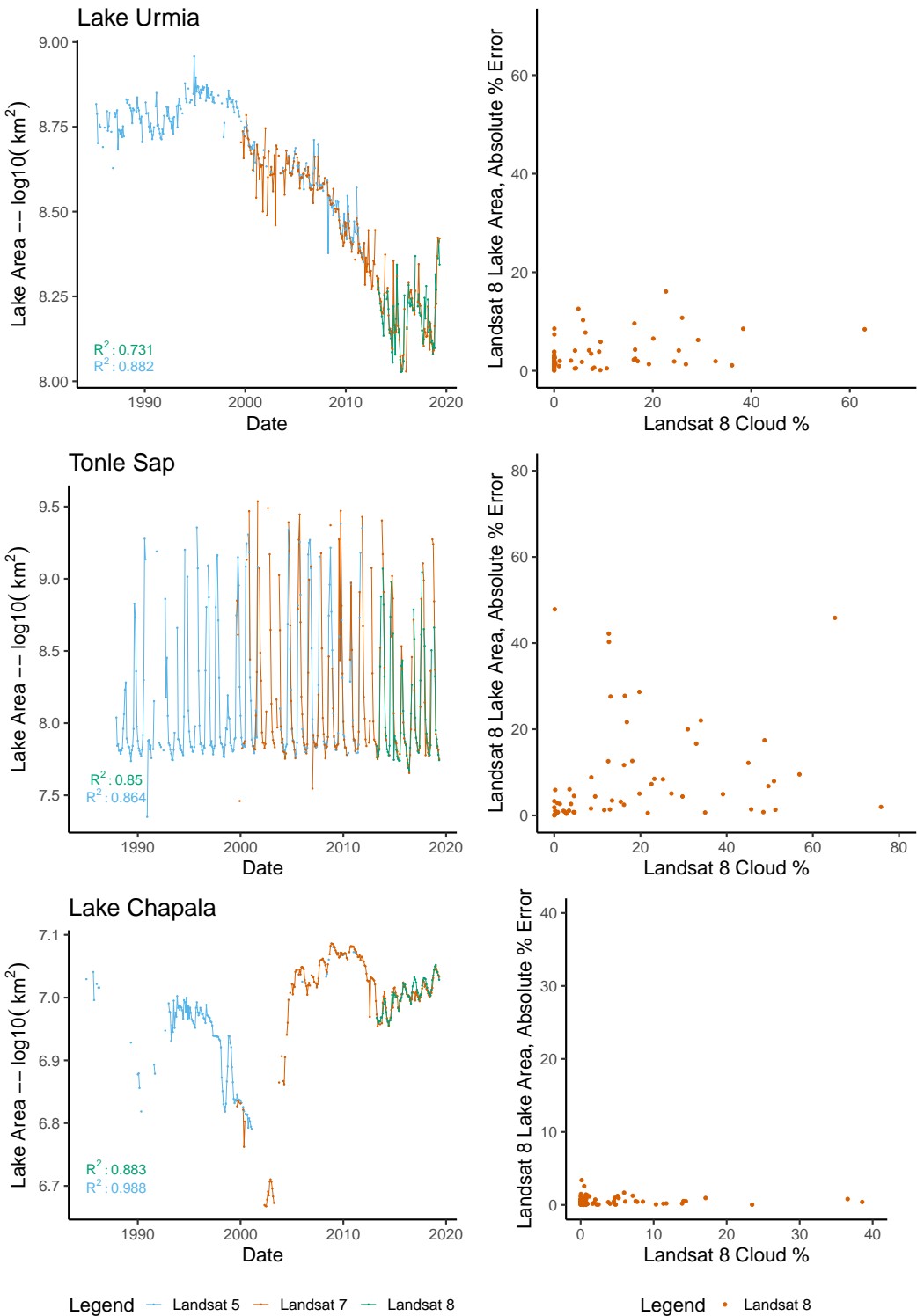

**Figure 5.** Implementation of the approach on lakes with documented changes. *Left*: Time series of monthly lake extent estimates from Landsat 5 (blue), Landsat 7 (brown) and Landsat 8 (green) for Lakes Urmia (*Top*), Tonle Sap (*Middle*), and Chapala (*Bottom*). *Right* Scatterplot of absolute percentage errors on Landsat 8 water extent estimates (compared to Landsat 7 estimates) against the proportion of the lake's maximum footprint that is covered by clouds.

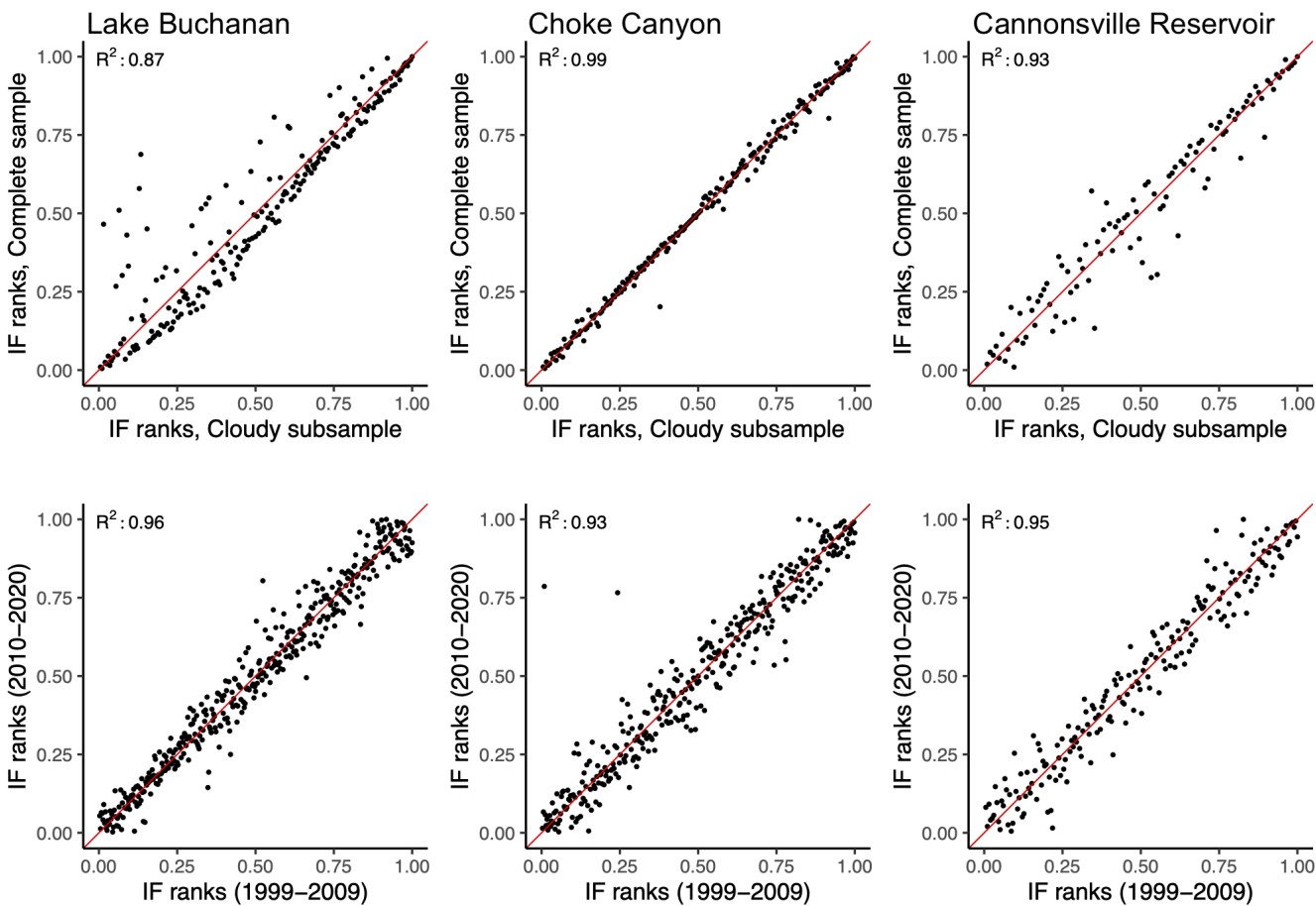

**Figure 6.** Assessment of the independence (*Top*) and stationarity (*Bottom*)assumptions for Lake Buchannan (*Left*), Choke Canyon Reservoir (*Middle*) and Cannonsville Reservoir (*Right*). *Top*: Inundation Frequency ranks per pixel estimated under cloudless condition (unsupervised classification, y-axis) plotted against corresponding ranks estimated using the full sample of observations (combined supervised-unsupervised classification, x-axis).*Bottom*: Inundation frequencies ranks per pixel estimated using the first (x-axis) and second (y-axis) half of the Landsat 7 observation period (1999-2019).

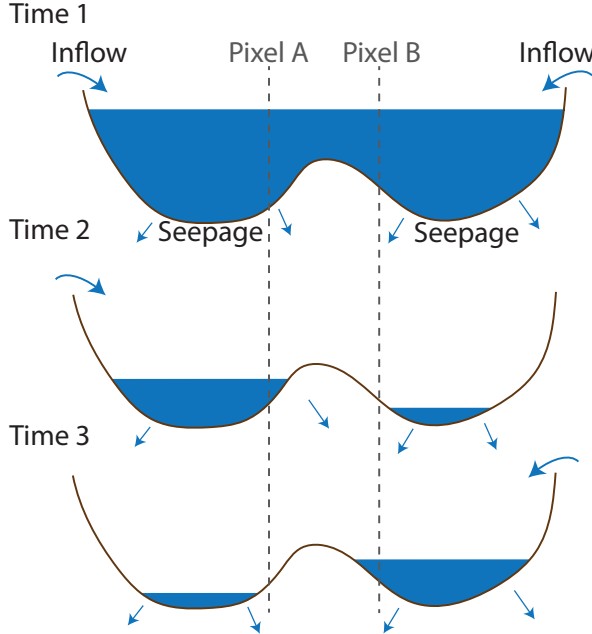

**Figure 7.** Violation of the homogeneity assumption in a fragmenting lake. A single body of water (top) might fragment into independent fragments when draining (middle and bottom). On the figure, the lake is drained by seepage and the two fragments are supplied by distinct tributaries. Under these conditions, pixels A and B might have an identical IF value but do not have an identical flooding status at times 2 and 3, hence violating the homogeneity assumption.