# Peer review of "A simple cloud-filling approach for remote sensing water cover"

_Hydrology and Earth System Sciences, 2020_

## Referee Comment (RC1) · Anonymous Referee #1 · 8 Jul 2020

General Comments

Mullen and Muller present a new method for producing time series of water extent in large, rapidly-changing and ecologically/culturally/economically important lakes. They use a novel approach implemented in Google Earth Engine (GEE) and validate their results against existing historical data, finding their method to work well, except when scenes contain snow/ice. Overall, the method is robust and the writing and figures in the manuscript are generally clear. However, I have several major concerns with the paper, chiefly related to the discussion of the method's limitations and the situation of this paper within the broader literature, described below. There are also several typos,

missing commas/parentheses and some incomplete sentences in the manuscript. I am not certain I caught all the errors, so I suggest the authors carefully edit the paper again before submitting a revised version.

Major Comments:

1. I would have thought that the specific cloud masking method could have a significant effect on the results, yet the cloud masking is only described in the SI and not given much attention in the manuscript. More discussion of the cloud masking method is needed in the main text. Furthermore, I would also suggest additional analysis and discussion about how the choice of a certain cloud-masking algorithm may or may not affect the results. For example, questions that I feel need to be addressed include what percent of pixels are cloudy/poor quality? How does this vary by lake/by year? Do lakes with greater cloudiness exhibit higher error than lakes with lower cloudiness?

2. The authors test their method over a small number of lakes – only 6 in total. But given the global availability of Landsat data, and the plethora of studies examining regional-to-global scale variability in surface water extent using Landsat/GEE (see comment #3), analyzing over only 6 lakes seems to me like a very small sample size. I encourage the authors to consider adding additional lakes to the analysis, perhaps with different environmental conditions such as in areas with high topography/high latitude (see comment #4). Relatedly, the authors should also consider adding discussion about the implementation of the method and the ease of running it – i.e. is the method computationally slow and therefore would be challenging to run over large areas or could this be reasonably run over, say, hundreds of large reservoirs?

3. This manuscript requires additional discussion of how this method fits in with the (very large) literature on monitoring lake extent using optical satellite imagery. The manuscript makes little mention of the work of Pekel et al. Nature, (2016), who map global variability in water extent using Landsat and GEE, or regional studies such as Zou et al. PNAS, (2018) or Wang et al., Nature Geoscience, (2018), or even the large

literature on reservoir monitoring using MODIS or other optical sensors (e.g. Gao et al., Water Resources Research, 2012). While I do appreciate that this method is designed to produce highly accurate time series for individual lakes which is different than the goals of many of these other studies, I feel more discussion is needed to distinguish specifically how this method is an advance compared to this previous work and particularly, what specific scientific questions this approach could answer that other approaches could not.

4. Relatedly, I also feel this manuscript is lacking some discussion about limitations and specific applications. The discussion about the different assumptions of the method is good; however, I was left wondering more specifically where this method might work and where it might fail. For example, would this method work in areas of high topography/high latitudes where topographic shadowing is an issue? What is the smallest lake this method would work on? Is there a relationship between cloudiness/size/error? I would also advise more discussion about what might have caused the outlier points removed in the time series analysis.

Specific Comments:

L1: "The empirical attribution of 'past' rapid hydrologic change"

L15: change "when applicable" to "where available"

L15: I would advise adding a sentence at the end of the abstract stating the importance/broader significance of your findings, instead of just stating that your method works

L18: In my opinion, the first few sentences of the paper are weak. I would suggest rewriting slightly (i.e. "Despite their importance, many lakes are undergoing rapid change..." makes little sense – the importance of lakes doesn't necessarily mean that lakes will not or should not undergo rapid change). Since "rapidly changing" is a key part of the manuscript, I would also suggest defining what you mean by rapid change

since the time scale implied by "rapid" can vary based on the reader's background.

L27: This sentence ("By providing") should start the next paragraph, not sit at the end of this one as it interrupts the flow

L31: The paragraph starts by talking about monitoring surface water extent, but then discusses radar altimeters before moving back to extent. I would suggest restructuring this paragraph, or at least the first sentence of it, as the current structure is confusing

L83: I suggest adding a sentence or two to the final paragraph of the introduction stating something like "we test this method over XX lakes, analyze its accuracy and demonstrate its utility" just to provide readers with a better road map for the manuscript

L86: I would call this first step something like "Masking" instead of pre-processing (see major comment #1 above).

Figure 1: I like this figure, but think it could be improved slightly by increasing the size of the image panels and decreasing the size of the arrows and white space. The image panels are hard to see in places and there's plenty of white space so it should be straightforward to make them a bit larger and easier to see.

L149: The sentence starting with "Indeed, visual inspection of satellite..." is unclear. I think what the authors are stating is that the area-elevation curves do not match the satellite-observed area, but this section could be clarified.

Figure 3: Please make the x and y labels and the symbols themselves much larger, it is nearly impossible to read the figure at this scale.

L181: "The analysis suggests..." this is not a complete sentence, please edit

Figure 5: Please make x and y labels larger

L229: change phrase starting with "if shadows..." to "shadows covering dry land in the vicinity of the lake may cause an overestimation of the surface area of the lake"

[Figure]

L225-234: Does the cloud masking method remove cloud shadows?

L225-234: Is the influence of topographic shadowing examined? Topographic shadowing, particularly in the NY lake (in winter) could influence classification accuracy (and would not be a randomly distributed error). Even if most of the lakes specifically examined here occur in the tropics or in areas with little-to-no surrounding topography, topographic shadowing issues would likely impact the applicability of this method in other areas and therefore should be discussed

---

## Referee Comment (RC2) · Anonymous Referee #2 · 8 Oct 2020

In this study, the authors developed a a hybrid remote sensing algorithm to estimate the time series of water surface areas over several lakes with rapid changes. The major novelty of this paper is that the proposed algorithm is still workable when the remote sensing images are partially covered by clouds/low-quality pixels. To enhance the capability of working under cloud condition, the authors first utilized unsupervised algorithm to classify pixels with high quality and then create a high-confidence inundation frequency (IF) image. Next, the supervised algorithm and the IF image were used to interpret masked pixels. The validation results against in situ observations indicate that the algorithm is able to monitor water surface changes with a high accuracy. This paper is well-written and easy to follow. However, there are several major concerns

related to the following aspects: (1) incomplete literature review (2) the applicability of the algorithm and (3) the design of the validation.

(1)Some efforts have been made to monitor surface water on global scale using remote sensing. It is not only necessary to acknowledge those work in this paper, but also meaningful to highlight the difference. For example, Pekel et al. (2016) developed a global water map by applying Machine Learning algorithm to Landsat images. Khandelwal et al. (2017) developed a method to monitor global lakes using MODIS images. Very similar to this paper, Zhao and Gao (2018) used an automatic approach to correct the contaminated images for assessment of reservoir surface area dynamics over 6,817 global reservoirs. They used the water occurrence map derived by Pekel et al. (2016) to correct the pixels with low quality, which is very close to the idea of IF image in this paper.

Ankush Khandelwal, Anuj Karpatne, Miriam E. Marlier, Jongyoun Kim, Dennis P. Lettenmaier, Vipin Kumar, An approach for global monitoring of surface water extent variations in reservoirs using MODIS data, Remote Sensing of Environment, Volume 202, 2017, Pages 113-128.

Pekel, J., Cottam, A., Gorelick, N. et al. High-resolution mapping of global surface water and its long-term changes. Nature 540, 418–422 (2016). https://doi.org/10.1038/nature20584.

Zhao, G., & Gao, H. (2018). Automatic correction of contaminated images forassessment of reservoir surface area dynamics. Geophysical Research Letters, 45, 6092–6099.

(2)The applicability of the algorithm needs to be further demonstrated given the fact only a few lakes were tested in this study. Besides this, all the selected lakes are very large considering the 30-m spatial resolution of Landsat. And the majority of the global lakes are much smaller than 1 km2. It would be interesting to add a section discussing the performance of the algorithm over small lakes. We can find the lowest

accuracy was obtained over the smallest lake in this study which raises a question - if the accuracy goes down when the size of lakes decreases.

(3)To validate the results, the authors compared all remotely sensed water extents with in situ observations. However, one of the highlights of the proposed approach is to interpret the masked pixels. So I am wondering how the algorithm performs during the cloudy season when lots of pixels are masked.

Other specific comments:

(1)Line24, miss a space in 'consequenceof'. (2)It would be better to increase the font size for all figures.

---

## Author Comment (AC1) · 3 Nov 2020

**Referee #1**

**General Comments**

Mullen and Muller present a new method for producing time series of water extent in large, rapidly-changing and ecologically/culturally/economically important lakes. They use a novel approach implemented in Google Earth Engine (GEE) and validate their results against existing historical data, finding their method to work well, except when scenes contain snow/ice. Overall, the method is robust and the writing and figures in the manuscript are generally clear. However, I have several major concerns with the paper, chiefly related to the discussion of the method's limitations and the situation of this paper within the broader literature, described below. There are also several typos, missing commas/parentheses and some incomplete sentences in the manuscript. I am not certain I caught all the errors, so I suggest the authors carefully edit the paper again before submitting a revised version.

**Response**
We thank the reviewer for their thorough and helpful review. We interpret the reviewers' comments as requiring (i) a more thorough discussion of the specific contribution of this manuscript in the context of the large literature on remotely sensed water detection, and (ii) a more careful analysis on the limitations of the approach. We address the reviewer's first point with a substantial rewrite of the introduction where we cast the proposed approach as addressing a gap filling challenge that is specific (and essential) to the type of high frequency historic reconstruction that we seek to achieve. We address the reviewer's second point by adding a set of numerical experiments exploring the sensitivity of the approach to different types of classification errors in the input data (i.e. clouds mistakenly classified as water or land, or water and land mistakenly classified as clouds), which we believe are the main limitation of our approach. We also added three lakes in the validation analysis to illustrate the application of the method in a small (~1km2) lake and in a mountainous/snowy setting.

We hope these new elements in the revised manuscript (specified in **red** below), and our specific responses (in **blue**) to their comments, adequately address the reviewer's concerns.

**Major Comments:**

1. I would have thought that the specific cloud masking method could have a significant effect on the results, yet the cloud masking is only described in the SI and not given much attention in the manuscript. More discussion of the cloud masking method is needed in the main text. Furthermore, I would also suggest additional analysis and discussion about how the choice of a certain cloud-masking algorithm may or may not affect the results. For example, questions that I feel need to be addressed include what percent of pixels are cloudy/poor quality? How does this vary by lake/by year? Do lakes with greater cloudiness exhibit higher error than lakes with lower cloudiness?

**Response**

We agree with the reviewer that cloud detection is an important prerequisite to our approach that merits further investigation. We will add a discussion section focusing on this dependence in the new manuscript, along with the following analyses:

- For each lake where in-situ validation data is available, we will add a scatterplot with the prediction error on the lake area plotted against the percentage of cloud cover. Results show that, across lakes, the prediction error does not increase significantly with cloudliness for the particular cloud masking algorithm that we use.

- We use numerical experiments to explore the sensitivity of our approach to the accuracy of cloud detection more generally. To investigate the effect of an overly eager cloud detection that tends to overestimate cloud cover, we changed a number of randomly selected (land and water) pixels of each image to cloud, and evaluate the method's ability to determine their original class (water or land). To investigate the effect of an algorithm that fails to fully capture (i.e., underestimates) clouds or cloud shadows, we "flipped" a number of randomly selected pixels from water to land (and vice versa) between the supervised and unsupervised classification steps of our algorithm. This emulates the fact that undetected cloud pixels might be misclassified as land or water. We then assess the effect of this misclassification on the method's ability to predict the class of clouded pixels. We find that our approach is robust to the former, but sensitive to the latter, type of error. The approach is therefore more compatible with an overly eager cloud detection algorithm, rather than an overly greedy one.

**Text modifications**
- Added scatterplots of errors vs. cloudliness for the validation lakes in Fig 3. These will replace the scatterplots currently in the middle column of the figure which are redundant with the reported R2 statistic.
- Added Figure and result section describing the numerical experiment.
- An added discussion section on the effect of "too little" information due to cloudliness, and overly eager cloud detection method or a smaller lake size.
- Modified conclusion section focusing on the practical implications of the method. We will specifically discuss the effect of an overly eager or greedy cloud detection algorithm.

2. The authors test their method over a small number of lakes – only 6 in total. But given the global availability of Landsat data, and the plethora of studies examining regional to-global scale variability in surface water extent using Landsat/GEE (see comment #3), analyzing over only 6 lakes seems to me like a very small sample size. I encourage the authors to consider adding additional lakes to the analysis, perhaps with different environmental conditions such as in areas with high topography/high latitude (see comment #4).

**Response**

While we agree with the reviewer that a larger sample size would be ideal, a persistent challenge that we ran into was to find reliable in situ observations of lake extents that matches

the monthly frequency and multi-decadal observation periods that we are targeting. Oftentimes we would find lake elevation time series but no reliable elevation-area relationships to obtain lake extents. Short of having a large sample of validation lakes, the revised manuscript uses a combination of targeted case studies and numerical experiments to investigate key limitations of our approach. Specifically, in response to comment #3 below, we now emphasize that our approach serves as a gap-filling algorithm, which purpose is to determine the class (water or land) of masked (cloud) pixels. We hypothesize that two main challenges can affect the accuracy of the gap-filling method: (i) too little information is provided in the input imagery or (ii) wrong information is contained in the input imagery or inundation frequency raster.

In the revised manuscript, we investigate these two limitations (too little information and wrong information) in two ways:

1. We use the two numerical experiments described above to investigate the effect of (i) too little (randomly masked pixels) and (ii) wrong (randomly flipped pixels) information. We add a third experiment where we add random noise to the inundation frequency raster. Comparing the outcome of the two latter experiments allows to differentiate the effect of having wrong information in the *dependent* (randomly flipped pixels) or in the *independent* (noise in the inundation frequency raster) data used to train the supervised classification.

2. We add three validation lakes to illustrate the effect of each limitation. Two small (~1km) lakes in Texas illustrate the effect of having too little information (i.e. a small number of available pixels). The other lake in upstate New York illustrates the effect of having wrong information, as the cold climate and mountainous terrain introduce errors in the unsupervised classification of clouds, water and land.

Both analyses suggest that the proposed method is more directly limited by wrong information, rather than too little information. This is consistent with the discussion on cloud masking of the previous comment. An overly eager cloud detection algorithm will err in favor of providing too little (rather than wrong) information and have a smaller effect on prediction performance.

**Text modifications**
- Three validation lakes added to Figure 3:
- An added figure and result section with the three proposed numerical experiments.
- An added discussion section on the effect of "too little" information due to cloudiness, and overly eager cloud detection method or a smaller lake size.
- We will modify the current discussion section focusing on water detection challenges to discuss the numerical experiment and focus on the drivers and effects of "wrong" information in the input classified imagery.

4b  Relatedly, the authors should also consider adding discussion about the implementation of the method and the ease of running it – i.e. is the method computationally slow and therefore would be challenging to run over large areas or could this be reasonably run over, say, hundreds of large reservoirs?

**Response**

With regards to implementation and scaling, we foresee no theoretical limit to the scalability of the proposed gap filling approach, thanks to Earth Engine's massive parallelization capability. The only "hard" limit lies in the need for users to define a region of interest (ROI)  over which to carry out the supervised classification. In the manuscript, these ROIs were determined manually by roughly following the maximum footprint of the considered lake. Methods can be developed to generate regions of interest automatically, for instance by creating a buffer around a certain threshold in our inundation frequency raster. However, we feel that developing and validating these approaches goes beyond the scope of this paper. We added a short discussion in the conclusion summarizing the above points. We also link to a working Earth Engine script for readers to evaluate for themselves the practicality of implementing the method.

**Text modification**
- Modified conclusion section focusing on the practical implications of the method. We will specifically discuss potential and limitations from a computational implementation perspective.

3. This manuscript requires additional discussion of how this method fits in with the (very large) literature on monitoring lake extent using optical satellite imagery. The manuscript makes little mention of the work of Pekel et al. Nature, (2016), who map global variability in water extent using Landsat and GEE, or regional studies such as Zou et al. PNAS, (2018) or Wang et al., Nature Geoscience, (2018), or even the large literature on reservoir monitoring using MODIS or other optical sensors (e.g. Gao et al., Water Resources Research, 2012). While I do appreciate that this method is designed to produce highly accurate time series for individual lakes which is different than the goals of many of these other studies, I feel more discussion is needed to distinguish specifically how this method is an advance compared to this previous work and particularly, what specific scientific questions this approach could answer that other approaches could not.

**Response**

The reviewer raises a good point and we edited the introduction to better describe the scope of the paper and its place with respect to previous work. In particular, we now reframe the general challenge of reconstructing historical time series of lake extent as a two-step problem. Only the second step is addressed by the proposed approach, although its sensitivity to errors from the first step are fully investigated in the revised manuscript (see comments 1 and 2 above).

1. The first step concerns the detection of land, water and clouds using multispectral imagery. This is a well studied problem that is generally well addressed, although well-known issues arise under specific circumstances. Improving the detection water, cloud and land from a

pixel's spectral signature is beyond the scope of this paper and we refer to the appropriate literature in the revised manuscript.

2. The second step of the problem is in essence a gap-filling challenge, where pixels classified as clouds during the first step must be reclassified as water or land. This problem is relatively easy to address if images are available at a short return time (e.g., daily MODIS imagery). For example, monthly water cover can be obtained from daily MODIS images, by simply masking clouded pixels from the analysis and taking the per-pixel mean inundation status of each stack of overlapping unmasked pixels. Similar approaches are used in several of the global studies mentioned by the reviewer (a detailed review of which is provided in the revised manuscript), where the main challenge lies in the classification of water, cloud and land (step 1), rather than the inference of the flooding status of cloudy pixels (step 2). Unfortunately, a similar approach cannot be applied to Landsat images due to their longer (two weeks) return time. Although Pekel's(2016) dataset is unique by providing monthly high resolution global water cover grids going back to the 1980's based on Landsat imagery, masked (cloudy) pixels are left unclassified and indistinguishable from missing pixels (e.g., due to Landsat 7 sensor failure or missing images). This leads to a significant underestimation of lake water extents (see Zhao and Gao 2018, GRL). The approach proposed in the manuscript seeks to address this type of situation and infer the classification status of mask landsat pixels. Its novelty, compared to previous work addressing the same issue, is its unique use of supervised classification to leverage historic inundation frequency (see response to comment #1 from reviewer 2).

**Text modification:**
Rewritten introduction to reflect the points made in our response above

4. Relatedly, I also feel this manuscript is lacking some discussion about limitations and specific applications. The discussion about the different assumptions of the method is good; however, I was left wondering more specifically where this method might work and where it might fail. For example, would this method work in areas of high topography/high latitudes where topographic shadowing is an issue? What is the smallest lake this method would work on? Is there a relationship between cloudiness/size/error? I would also advise more discussion about what might have caused the outlier points removed in the time series analysis.

**Response**
We thank the reviewer for their comment and hope that the new discussion section about the approach's sensitivity to classification errors in the input imagery (comments 1-2 above) will address their concerns. Many of the questions asked by the reviewer in their comment boil down to the effect of pixel misclassification (or, more fundamentally, to too little or wrong information) on the method's ability to re-classify as land or water the pixels that were previously identified as clouds. The new analyses in the revised manuscript (comments 1-2 above) show that the method is robust to the former (too little information) but sensitive to the latter (wrong information).

In practice, this means that the method will not perform well in locations where circumstances (topographic shading, cloud shading, snow/ice, etc) makes it difficult to reliably distinguish water from clouds and land using multispectral imagery. The inability to properly classify clouds, water and land in the input imagery gives rise to the outliers that are subsequently removed from the time series analysis. This point is clarified in the revised manuscript when discussing the numerical experiments and the added lake from upstate New York.

In contrast, the method is likely to be generally robust to application to smaller lake sizes, despite a limited number of unmasked pixels available to train the supervised classification. However, performance is contingent on the assumptions that (i) the location of cloud and water pixels are statistically independent and (ii) that lake bathymetry does not change over time. We posit in the original manuscript that these assumptions are less likely to be satisfied in small lakes, but in situ data is insufficient to formally test this hypothesis. The two assumptions appear to be satisfied for the ~1km2 lakes that we added to the validation sample in the revised manuscript.

**Text modification:**
- Added figure and results section on the numerical experiments
- Added discussion section on the effect of "too little" information due to cloudiness, and overly eager cloud detection method or a smaller lake size.
- We will modify the current discussion section focusing on water detection challenges to discuss the numerical experiment and focus on the drivers and effects of "wrong" information in the input classified imagery.
- Modified conclusion section focusing on the practical implications of the method. We will specifically discuss accurate water detection in unmasked pixels as a key prerequisite to the method.

Specific Comments:
**Response**
We thank the reviewer for their detailed specific comments, which we think generally improve the readability of the manuscript. We will address all comments in the revised manuscript, provided that they still apply (i.e. if the referred text was not already modified to address a major comment).

L1: "The empirical attribution of 'past' rapid hydrologic change" L15: change "when applicable" to "where available"

L15: I would advise adding a sentence at the end of the abstract stating the importance/broader significance of your findings, instead of just stating that your method works

L18: In my opinion, the first few sentences of the paper are weak. I would suggest rewriting slightly (i.e. "Despite their importance, many lakes are undergoing rapid change. . ." makes little sense – the importance of lakes doesn't necessarily mean that lakes will not or should not

undergo rapid change). Since "rapidly changing" is a key part of the manuscript, I would also suggest defining what you mean by rapid change paper since the time scale implied by "rapid" can vary based on the reader's background.

L27: This sentence ("By providing") should start the next paragraph, not sit at the end of this one as it interrupts the flow

L31: The paragraph starts by talking about monitoring surface water extent, but then discusses radar altimeters before moving back to extent. I would suggest restructuring this paragraph, or at least the first sentence of it, as the current structure is confusing

L83: I suggest adding a sentence or two to the final paragraph of the introduction stating something like "we test this method over XX lakes, analyze its accuracy and demonstrate its utility" just to provide readers with a better road map for the manuscript

L86: I would call this first step something like "Masking" instead of pre-processing (see major comment #1 above).

Figure 1: I like this figure, but think it could be improved slightly by increasing the size of the image panels and decreasing the size of the arrows and white space. The image panels are hard to see in places and there's plenty of white space so it should be straightforward to make them a bit larger and easier to see.

L149: The sentence starting with "Indeed, visual inspection of satellite. . ." is unclear. I think what the authors are stating is that the area-elevation curves do not match the satellite-observed area, but this section could be clarified.

Figure 3: Please make the x and y labels and the symbols themselves much larger, it is nearly impossible to read the figure at this scale.

L181: "The analysis suggests. . ." this is not a complete sentence, please edit Figure 5: Please make x and y labels larger

L229: change phrase starting with "if shadows. . ." to "shadows covering dry land in the vicinity of the lake may cause an overestimation of the surface area of the lake"

L225-234: Does the cloud masking method remove cloud shadows?

L225-234: Is the influence of topographic shadowing examined? Topographic shadowing, particularly in the NY lake (in winter) could influence classification accuracy (and would not be a randomly distributed error). Even if most of the lakes specifically examined here occur in the tropics or in areas with little-to-no surrounding topography, topographic shadowing issues would likely impact the applicability of this method in other areas and therefore should be discussed

**References**

Avisse, Nicolas, et al. "Monitoring small reservoirs' storage with satellite remote sensing in inaccessible areas." *Hydrology and Earth System Sciences* 21.12 (2017): 6445.

Ankush Khandelwal, Anuj Karpatne, Miriam E. Marlier, Jongyoun Kim, Dennis P. Lettenmaier, Vipin Kumar, An approach for global monitoring of surface water extent variations in reservoirs using MODIS data, Remote Sensing of Environment, Volume 202, 2017, Pages 113-128.

Pekel, J., Cottam, A., Gorelick, N. et al. High-resolution mapping of global surface water and its long-term changes. Nature 540, 418–422 (2016). https://doi.org/10.1038/nature20584.

Zhao, G., & Gao, H. (2018). Automatic correction of contaminated images for assessment of reservoir surface area dynamics. Geophysical Research Letters, 45, 6092– 6099.

---

## Author Comment (AC2) · 3 Nov 2020

**Referee #2**

In this study, the authors developed a hybrid remote sensing algorithm to estimate the time series of water surface areas over several lakes with rapid changes. The major novelty of this paper is that the proposed algorithm is still workable when the remote sensing images are partially covered by clouds/low-quality pixels. To enhance the capability of working under cloud condition, the authors first utilized unsupervised algorithm to classify pixels with high quality and then create a high-confidence inundation frequency (IF) image. Next, the supervised algorithm and the IF image were used to interpret masked pixels. The validation results against in situ observations indicate that the algorithm is able to monitor water surface changes with a high accuracy. This paper is well-written and easy to follow. However, there are several major concerns related to the following aspects: (1) incomplete literature review (2) the applicability of the algorithm and (3) the design of the validation.

**Response**
We thank the reviewer for the positive comments and hope that our responses (in **blue**) to their comments below, and the associated modifications to the manuscript (in **red**) have addressed their three specific concerns.

(1)Some efforts have been made to monitor surface water on global scale using remote sensing. It is not only necessary to acknowledge those work in this paper, but also meaningful to highlight the difference. For example, Pekel et al. (2016) developed a global water map by applying Machine Learning algorithm to Landsat images. Khandelwal et al. (2017) developed a method to monitor global lakes using MODIS images. Very similar to this paper, Zhao and Gao (2018) used an automatic approach to correct the contaminated images for assessment of reservoir surface area dynamics over 6,817 global reservoirs. They used the water occurrence map derived by Pekel et al. (2016) to correct the pixels with low quality, which is very close to the idea of IF image in this paper.

**Response**
We thank the reviewer for their suggestions and have rewritten the introduction to address it (also in response to comment 3 from reviewer 1). In the new introduction we provide an extensive review of the literature (now specifically referring to the mentioned studies, among others), and clarify our contribution as specifically addressing the problem of retrieving classification information from masked pixels. We agree with the reviewer that the approach from Zhao and Gao (Z&G), which we were not aware of, addresses the same problem with a comparable method, although with important differences. Z&G is indeed similar to the proposed approach in that it leverages historical inundation frequency information to infer the flooding status of masked pixels. Both approaches rely on the assumption that all masked pixels with inundation frequency above a certain threshold should be designated as flooded. However, the two approaches differ in the way they define this threshold, which varies from image to image depending on the current level of the lake. Z&G use a heuristic approach based on the shape of the histogram of unmasked flooded pixels to determine the threshold. In contrast, our approach

relies on machine learning (supervised classification) to leverage the relationship between the inundation frequency and current status of *all* unmasked pixels: flooded and unflooded. By relying on statistical relationships embedded within each image, rather than on arbitrary heuristic rules that are not able to vary across images, the proposed approach is more flexible and less reliant on user input. We now explicitly refer to Z&G in the introduction, along with the other approaches that we are aware of that address the same challenge (e.g., Avisse et al 2017 and Khandelwal et al 2017, which both rely on a DEM), and clarify the specific contribution of the proposed approach.

**Text modification:**
Rewritten introduction to reflect the points made in our response above

2)The applicability of the algorithm needs to be further demonstrated given the fact only a few lakes were tested in this study. Besides this, all the selected lakes are very large considering the 30-m spatial resolution of Landsat. And the majority of the global lakes are much smaller than 1 km2. It would be interesting to add a section discussing the performance of the algorithm over small lakes. We can find the lowest accuracy was obtained over the smallest lake in this study which raises a question - if the accuracy goes down when the size of lakes decreases.

**Response**
We agree with the reviewer and address lake size in a revised conclusion section focused on practical implications. More generally, the validation of our approach (now) focuses on investigating two key potential limitations: its sensitivity to (i) too little information in the input imagery and (ii) wrong information in the input classified imagery, both of which are now discussed in the discussion section. We use a combination of numerical experiments and targeted case studies to assess both limitations, as described in our response to 'comment 2' from reviewer 1.

We make the argument that lake size mainly affects the performance of our approach by decreasing the amount of information available to classify masked pixels. This effect is investigated in the revised manuscript by running a numerical experiment where a predetermined fraction of randomly selected pixels were masked from each input image. We find that the method's ability to predict the class (water or land) of the artificially masked images is robust to an increasingly large fraction of masked images. We interpret this as indicative of the method's applicability to small lakes, provided that (i) the location of cloud and water pixels are statistically independent and (ii) that lake bathymetry does not change over time. We posit in the original manuscript that these assumptions are less likely to be satisfied in small lakes, but in situ data is insufficient to formally test this hypothesis. The two assumptions appear to be nonetheless satisfied for the two ~1km2 lakes that we added to the validation sample in the revised manuscript. (Unfortunately, while most global lakes are indeed smaller than 1km2, very few have openly available in-situ data of their surface area at the frequency and coverage period considered in this study.)

In contrast, the method was very sensitive to classification errors in the input imagery (e.g., clouds misclassified as water, water misclassified as land, etc), which we interpret as indicative of its limitation in situations where circumstances (topography, climate) makes it difficult to systematically identify water on the input multispectral imagery. These effects might have confounded the effect of lake size in the original manuscript: lakes in NY state, where topography and climate made water detection more challenging, were also generally smaller than the other validation lakes in Texas. To disentangle these effects, we add three lakes in the revised manuscript: two small lakes (~1km2) in Texas where water detection is straightforward, and one larger lake in NY where water detection is more challenging. We find that the method performs well on the former and poorly on the latter. This is in line with the numerical experiment results and suggests that water detection, rather than lake size, stands as a more immediate limitation of the approach.

**Text modification:**
- Added Figure and results section on the numerical experiments
- Added validation lakes in Figure 3
- Added discussion section on the effect of "too little" information due to cloudiness, an overly eager cloud detection method or a smaller lake size.
- We will modify the current discussion section focusing on water detection challenges to discuss the numerical experiment and focus on the drivers and effects of "wrong" information in the input classified imagery.
- Modified conclusion section focusing on the practical implications of the method. We will specifically discuss the potential and limitations of the method's application to small lakes.

(3)To validate the results, the authors compared all remotely sensed water extents with in situ observations. However, one of the highlights of the proposed approach is to interpret the masked pixels. So I am wondering how the algorithm performs during the cloudy season when lots of pixels are masked.

**Response**
We thank the reviewer for their suggestion, and have added to the revised manuscript a figure for each validation lake representing lake extent estimation error against cloudiness. We find that errors are not significantly correlated to the extent of cloud cover if clouds are appropriately detected., This finding is in line with our interpretation that the approach is robust to having too little (small or heavily clouded lakes), information in the input imagery. However, our numerical experiment also indicates that errors are correlated to the prevalence of *wrong* information (misclassification of water or land) in the input imagery. This indicates that overly greedy cloud/shadow masking algorithms will yield better results compared with algorithms that underpredict cloud/shadow pixels. This also entails that challenges will arise when many cloud pixels are difficult to detect (e.g., waspy clouds or cloud edges).

**Text modification**
- Added scatterplots of errors vs. cloudliness for the validation lakes in Fig 3. These will replace the scatterplots currently in the middle column of the figure which are redundant with the reported R2 statistic.
- Added discussion section on the effect of "too little" information due to cloudiness, an overly eager cloud detection method or a smaller lake size.
- Modified conclusion section focusing on the practical implications of the method. We will specifically discuss the potential and limitations of the method's application to lakes with high/frequent cloud cover.

Other specific comments:
Thank you for your comments, which we will be sure to address in the revised manuscript.

(1) Line24, miss a space in 'consequenceof'.
(2) It would be better to increase the font size for all figures.

References

Avisse, Nicolas, et al. "Monitoring small reservoirs' storage with satellite remote sensing in inaccessible areas." *Hydrology and Earth System Sciences* 21.12 (2017): 6445.

Ankush Khandelwal, Anuj Karpatne, Miriam E. Marlier, Jongyoun Kim, Dennis P. Lettenmaier, Vipin Kumar, An approach for global monitoring of surface water extent variations in reservoirs using MODIS data, Remote Sensing of Environment, Volume 202, 2017, Pages 113-128.

Pekel, J., Cottam, A., Gorelick, N. et al. High-resolution mapping of global surface water and its long-term changes. Nature 540, 418–422 (2016). https://doi.org/10.1038/nature20584.

Zhao, G., & Gao, H. (2018). Automatic correction of contaminated images for assessment of reservoir surface area dynamics. Geophysical Research Letters, 45, 6092– 6099.

---

## Author Response (AR1)

Dear Dr Kelleher,

Please find attached the revised version of our previously submitted manuscript: "Assessing historic water extents in rapidly changing lakes: a hybrid remote sensing classification approach", now renamed "**A simple cloud-filling approach for remote sensing water cover assessments**", for consideration as a research article at HESS.

We have carefully addressed the concerns voiced by yourself and the two reviewers and are grateful for the opportunity to resubmit. The most substantive changes to the manuscript are:

1. A complete rewrite of the introduction to re-scope the paper as proposing a new simple cloud-filling approach for the remote sensing assessment of water cover. We have also revised the title of the paper to reflect this new scope. The revised introduction now contains an extensive literature review that clearly outlines the gap that this study seeks to fill. In response to your comment, we now also explicitly articulate the specific type of new insights that the proposed method will allow to provide.

2. An enhanced validation of the approach, which now relies on numerical experiments and three added validation lakes. These additional analyses focus on the four fundamental assumptions of the approach and allow us to rigorously assess its key limitations. In response to your comment, these limitations are explicitly discussed in the discussion and conclusion sections of the revised manuscript.

We are extremely grateful to both reviewers and to yourself for taking the time to read our paper and for offering such constructive comments. We believe that the revised manuscript has been substantially improved thanks to this feedback and hope that we have addressed all reviewers' concerns. A point-by-point response (in blue) to all reviewers' comments and the related revised text (in red) are provided below. A track-change version of the manuscript and SI is enclosed.

Thanks again for your consideration.

Sincerely,

Marc Muller, on behalf of all authors.

**Editor:**

Dear authors,

The manuscript has received two constructive reviews. Both reviewers note that the approach proposed in this manuscript is of interest to the broader remote sensing community. However, both reviewers had recommendations that should be addressed in an updated manuscript.

In particular, both reviewers had common concerns:

-That key literature was missing from the manuscript, necessitating placing this work within the context of this broader literature on surface water extents: Both reviewers note that key literature is missing.

While your response to the reviewers indicates that you will use this literature to revise your introduction, I would encourage you to go beyond this. In particular, I would encourage bringing this literature into the discussion section, to compare and contrast your approach with these existing approaches. Placing your findings in the context of this existing literature within the discussion section will help to show what is novel about this approach and how it advances the methodology of detecting lake extents.

The addition to the discussion that is most important will be clearly articulating how this approach enables researchers to answer new questions about lake water extent – what does this new approach allow that other approaches do not? As noted by Reviewer 1, what new insights can be gained? Crafting this novelty within the discussion section by bringing in the broad body of literature on lake water extent detection is necessary to show how this manuscript goes beyond what has already been accomplished in this research area.

**Response**

Thank you for your thoughtful comments and suggestions. We agree that the manuscript was in need of a stronger discussion about the specific contribution of the proposed method and have completely rewritten the introduction to reflect that. We made the stylistic choice of discussing the novelty of the approach in the introduction, rather than the discussion, because we believe that it is critical that the specific contributions of the paper be made clear upfront: as a reader why go through the trouble of even reading the paper if one is not convinced that what is presented is new and useful?

Specifically, we start by making the case that gap-filling approaches are necessary to leverage landsat-derived information to attribute historical change (L1-67). We then review the (few) gap-filling approaches that already exist to infer the binary status (e.g., wet vs dry) of masked pixels. We argue that they either (i) require ancillary data, (ii) rely on arbitrary heuristics, or (iii) are not straightforward to implement and scale (L68-84). We then present our approach as addressing all three issues simultaneously by using a standard machine learning algorithm to fully leverage the information embedded in the masked input dataset (L 85-94).

In terms of new insights, this allows our approach to be robust to detection errors, which we now formally test using numerical experiments. It also allows it to be fully compatible with Google Earth Engine with the scalability and portability that this implies. This last point represents an important contribution: the

approach is packaged into a relatively simple (~10 lines) javascript function that can be easily integrated into any google earth engine script.

**Text Modifications**

**L60-67:**
Introduction

[revised manuscript text omitted]

-That only a few lakes have been chosen for analysis and validation: I like that the dataset used for testing this approach draws on lakes of different sizes and with different data availability, and in different places. I also appreciate that you have already conducted additional analysis and plan to include three additional lakes within the revised manuscript.

That said, I do think that the limited dataset, as well as your experiences searching for validation data, should be acknowledged within the discussion section. In particular, reviewer 1 asks for more discussion of the limitations of the approach – including where it will work, and how it will be impacted by other environmental conditions. Given the size of the training dataset, discussing these other potential limitations will add to the contribution of this work to the broader literature. When revising this Discussion section, I would encourage you to broadly discuss challenges to the satellite detection of lake water extents, and then move into more specific challenges for your application. Adding a more general discussion will help place this work in the context of other similar studies, and provide more context on common challenges to these types of approaches.

**Response**
Thank you for your comment and understanding regarding the data availability challenge for in-situ validation. To more rigorously evaluate our approach, we have re-thought our validation as a two-step process specifically focusing on the four fundamental assumptions of the algorithm (L 126-145). In the first step, we investigate how sensitive the gap filling algorithm is to deviations from each assumption using numerical experiments (described in L156-195). This analysis is now presented as a main result in the revised manuscript (L. 243-274). In the second step we apply the approach to monitor the extent of 9 specific lakes that span a variety of sizes, topographies, climates and levels of data availability. We use these case studies to discuss the propensity for deviations from the four fundamental assumptions to emerge in practice (L 306-359).

**Text modifications:**
**L 126-144:**

2.1 Gap filling algorithm

[revised manuscript text omitted]

-That there may be more limitations to the approach – due to environmental conditions – than discussed in the current manuscript: Again, I would encourage a thoughtful discussion of environmental conditions that may limit this approach, taking into account many of the points raised by the reviewers (e.g., topographic shading).

**Response**
Thanks for your comment. We now explicitly discuss the limitation of the approach in the discussion (L 326-329 and 359-363) and conclusion (L 371-381) sections, based on our analysis of the four

fundamental assumptions of the approach. Specifically, two key limitations stand out. First, detection errors in the input imagery are the most salient limitation of the approach and include the cloud detection and topographic shading issues described by the reviewers. We assess the relative effect of different types of detection errors and provide practical avenues to mitigate them (e.g., erring towards an excessively eager cloud detector). Second the method requires the a-priori delineation of regions where the relationship between the inundation frequency and inundation status of pixels is homogeneous. This condition is trivial to satisfy for individual lakes but may limit the scalability of the approach for hydrologically complex regions (e.g., fragmenting lakes and complex wetlandscapes).

**Text Modifications**
**L326-329**
These results illustrate a key limitation of the approach, that gap-filling accuracy is constrained by the accuracy of the input ternary imagery. They also suggest that the approach is more compatible with an overly eager cloud detector: by overestimating cloud cover the input imagery will err in favor of providing too little (rather than wrong) information, which has a smaller effect on the accuracy of the gap filling algorithm.

**L359-363**
This assumption clearly holds for the non-disjoint bodies of water that are considered in this study, but may not apply to bodies of water that fragment upon drainage (Figure 7). There, the gap filling algorithm should be applied independently for each homogeneous region. The need to identify homogeneous regions *a priori* in fragmenting lakes and more complex wetlandscapes is an important limitation of the approach.

**L371-381**
However, the analyses also outlined two important limitations of the approach. First, the approach is sensitive to classification errors in the input imagery, particularly in small lakes. Misclassification of the output binary classes (here wet/dry) have a stronger impact on performance than misidentification of masked pixels (here clouds) and the effect is exacerbated when unmasked lakes pixels fall below 25 ha (roughly 17 x 17 Landsat pixels). This further implies that the approach might not perform well in locations where circumstances (topographic shading, cloud shading, snow/ice, etc) makes it difficult to reliably distinguish water from clouds and land using multispectral imagery. In contrast, the method appears generally robust to situations where a limited number of input classified pixels are available for training (e.g., small lakes or high cloud coverage). These two observations imply that the approach is preferably combined with a cloud detector that tends to overestimate cloud coverage. Second, the approach requires the *a priori* identification of homogeneous regions, where the relationship between the inundation frequency and inundation status of pixels is unique. This requirement limits the scalability of the approach in complex wetlandscapes, where the relationship might vary through space.

Both reviewers also had a few minor comments that would improve the manuscript. In particular, the introduction is missing a scope in the final paragraph (noted by Reviewer 1). In addition, the figures were challenging to read, particularly axis labels, legends, and symbols. I encourage the authors to revisit all figures to ensure that they are readable.

**Response**
Thanks for your comment. We revised the figures to make the text legible and rewrote the introduction in a way that (we hope) clearly states the scope of the paper.

I request that the authors incorporate their proposed revisions into the discussion paper for re-review.

**Referee #1**

**General Comments**

Mullen and Muller present a new method for producing time series of water extent in large, rapidly-changing and ecologically/culturally/economically important lakes. They use a novel approach implemented in Google Earth Engine (GEE) and validate their results against existing historical data, finding their method to work well, except when scenes contain snow/ice. Overall, the method is robust and the writing and figures in the manuscript are generally clear. However, I have several major concerns with the paper, chiefly related to the discussion of the method's limitations and the situation of this paper within the broader literature, described below. There are also several typos, missing commas/parentheses and some incomplete sentences in the manuscript. I am not certain I caught all the errors, so I suggest the authors carefully edit the paper again before submitting a revised version.

**Response**
We thank the reviewer for their thorough and helpful review. We interpret the reviewers' comments as requiring (i) a more thorough discussion of the specific contribution of this manuscript in the context of the large literature on remotely sensed water detection, and (ii) a more careful analysis on the limitations of the approach. We address the reviewer's first point with a substantial rewrite of the introduction where we cast the proposed approach as addressing a gap filling challenge that is specific (and essential) to the type of high frequency historic reconstruction that we seek to achieve. We address the reviewer's second point by adding a set of numerical experiments exploring the sensitivity of the approach to different types of classification errors in the input data (i.e. pixels misclassified classified as water or land, or water and land pixels mistakenly classified as clouds), which we believe are the main limitation of our approach. We also added three lakes in the validation analysis to illustrate the application of the method in a small (~1km2) lake and in a mountainous/snowy setting.

**Major Comments:**

1. I would have thought that the specific cloud masking method could have a significant effect on the results, yet the cloud masking is only described in the SI and not given much attention in the manuscript. More discussion of the cloud masking method is needed in the main text. Furthermore, I would also suggest additional analysis and discussion about how the choice of a certain cloud-masking algorithm may or may not affect the results. For example, questions that I feel need to be addressed include what percent of pixels are cloudy/poor quality? How does this vary by lake/by year? Do lakes with greater cloudiness exhibit higher error than lakes with lower cloudiness?

**Response**
We agree with the reviewer that cloud detection is an important prerequisite to our approach that merits further investigation. Our revised validation approach focuses on the four fundamental assumptions of the algorithm: input classification accuracy, stationarity, independence and heterogeneity (see L 125-144 of the revised manuscript). Questions associated with cloud cover and cloud detection mentioned in your comment relate to the first assumption by specifically affecting the amount of information fed into the supervised classifier. The revised discussion section focuses in part on this dependence and specifically discussed the effect of cloud detection on performance (L 315-331). This discussion is supported by two added analyses:

- Empirical analysis: For each lake, we added a scatterplot with the prediction error on the lake area plotted against the percentage of cloud cover. Results show that, across lakes, the prediction error does not increase significantly with cloudiness for the particular cloud masking algorithm that we use.

- Numerical experiment: We use numerical experiments to explore the sensitivity of our approach to the accuracy of cloud detection more generally. To investigate the effect of an overly eager cloud detection that tends to overestimate cloud cover, we changed a number of randomly selected (land and water) pixels of each image to cloud, and evaluate the method's ability to determine their original class (water or land). To investigate the effect of an algorithm that fails to fully capture (i.e., underestimates) clouds or cloud shadows, we "flipped" a number of randomly selected pixels from water to land (and vice versa) between the supervised and unsupervised classification steps of our algorithm. This emulates the fact that undetected cloud (or shadow) pixels might be misclassified as land or water. We then assess the effect of this misclassification on the method's ability to predict the class of clouded pixels. We find that our approach is robust to the former but sensitive to the latter type of error. The approach is therefore more compatible with an overly eager cloud detection algorithm, rather than an overly cautious one.

**Text modifications**
- Added scatterplots of errors vs. cloudiness for the validation lakes on Figs 4, 5 and S1 of the revised manuscript.
- Added Figure 3 and Sections 2.2 and 3.1 describing the numerical experiments and their results.
- An added discussion (Section 4) on the effect of classification errors and cloud detection on the gap-filling accuracy.
- Modified conclusion (Section 5) to explicitly discuss the practical limitations of the method, including with regard to cloud cover and detection.

2. The authors test their method over a small number of lakes – only 6 in total. But given the global availability of Landsat data, and the plethora of studies examining regional to-global scale variability in surface water extent using Landsat/GEE (see comment #3), analyzing over only 6 lakes seems to me like a very small sample size. I encourage the authors to consider adding additional lakes to the analysis, perhaps with different environmental conditions such as in areas with high topography/high latitude (see comment #4).

**Response**

While we agree with the reviewer that a larger sample size would be ideal, a persistent challenge that we ran into was to find reliable in situ observations of lake extents that matches the monthly frequency and multi-decadal observation periods that we are targeting. Oftentimes we would find lake elevation time series but no reliable elevation-area relationships to obtain lake extents. Short of having a large sample of validation lakes, the revised manuscript uses a combination of targeted case studies and numerical experiments to investigate key limitations of our approach. Specifically, in response to comment #3 below, we now emphasize that our approach serves as a gap-filling algorithm, which purpose is to determine the class (water or land) of masked (cloud) pixels. We hypothesize that two main issues associated with the accuracy of the input classified imagery can affect the accuracy of the gap-filling method: (i) too little information is provided in the input imagery or (ii) wrong information is contained in the input imagery or inundation frequency raster.

In the revised manuscript, we investigate these two limitations (too little information and wrong information) in two ways:

1. We use the numerical experiments described above to investigate the effect of (i) too little (randomly masked pixels) and (ii) wrong (randomly flipped pixels) information.
2. We add three validation lakes to illustrate the effect of each limitation. Two small (~1km) lakes in Texas illustrate the effect of having too little information (i.e. a small number of available pixels). The other lake in upstate New York illustrates the effect of having wrong information, as the cold climate and mountainous terrain introduce errors in the unsupervised classification of clouds, water and land.

Both analyses suggest that the proposed method is more directly limited by wrong information, rather than too little information. This is consistent with the discussion on cloud masking of the previous comment. An overly eager cloud detection algorithm will err in favor of providing too little (rather than wrong) information and have a smaller effect on prediction performance.

**Text modifications**
- Three validation lakes added to Figures 4 and S1.
- An added Figure 3 and methods (Section 2.2) and result (Section 3.1) sections describing the numerical experiments and their result.
- Added Figure 3 and Sections 2.2 and 3.1 describing the numerical experiments and their results.
- An added discussion (Section 4) on the effect of classification errors and cloud detection on the gap-filling accuracy.

4b Relatedly, the authors should also consider adding discussion about the implementation of the method and the ease of running it – i.e. is the method computationally slow and therefore would be challenging to run over large areas or could this be reasonably run over, say, hundreds of large reservoirs?

**Response**

With regards to implementation and scaling, the "simplicity" of the gap-filling approach and its reliance on readily available machine learning techniques makes it fully compatible with Google Earth Engine. We see this as one of the key advantages of the approach, which can then leverage the scalability and portability of the Earth Engine platform. However, an important limit to scalability is the need to comply with the homogeneity assumption of the approach (see L191-199). This requires the ex ante identification of regions where the relationship between the historical inundation frequency and their current flooding status is homogeneous. While not a problem for the monitoring of distinct lakes, this requirement might be challenging to satisfy in hydrologically complex landscape such as wetland and water bodies that fragment when being drained. This limitation is now extensively discussed in the Discussion (Section 4, L353-359) and Conclusion (Section 5, L374-376) sections. We also added permanent links to working Earth Engine scripts of all analyses for readers to evaluate for themselves the practicality of implementing the method (L382-385).

**Text modifications**
- Added Sections 2.2 and 3.1 describing a numerical experiment to assess the robustness of the approach to violations of the homogeneity assumption.
- Modified Sections 4 and 5 to discuss practical cases of violations of the homogeneity assumptions and implications for the scalability of the approach.

3. This manuscript requires additional discussion of how this method fits in with the (very large) literature on monitoring lake extent using optical satellite imagery. The manuscript makes little mention of the work of Pekel et al. Nature, (2016), who map global variability in water extent using Landsat and GEE, or regional studies such as Zou et al. PNAS, (2018) or Wang et al., Nature Geoscience, (2018), or even the large literature on reservoir monitoring using MODIS or other optical sensors (e.g. Gao et al., Water Resources Research, 2012). While I do appreciate that this method is designed to produce highly accurate time series for individual lakes which is different than the goals of many of these other studies, I feel more discussion is needed to distinguish specifically how this method is an advance compared to this previous work and particularly, what specific scientific questions this approach could answer that other approaches could not.

**Response**
The reviewer raises a good point and we edited the introduction to better describe the scope of the paper and its place with respect to previous work. In particular, we now reframe the general challenge of reconstructing historical time series of lake extent as a two-step problem. Only the second step is addressed by the proposed approach, although its sensitivity to errors from the first step are fully investigated in the revised manuscript (see comments 1 and 2 above).

1. The first step concerns the detection of land, water and clouds using multispectral imagery. This is a well studied problem that is generally well addressed, although well-known issues arise under specific circumstances. Improving the detection water, cloud and land from a pixel's spectral signature is beyond the scope of this paper and we refer to the appropriate literature in the revised manuscript.

2. The second step of the problem is in essence a gap-filling challenge, where pixels classified as clouds during the first step must be reclassified as water or land. Pekel's(2016) dataset is unique by

providing monthly high resolution global water cover grids going back to the 1980's based on Landsat imagery, thus addressing step 1 above. However, masked (cloudy) pixels are left unclassified, leading to a significant underestimation of lake water extents (see Zhao and Gao 2018, GRL). The approach proposed in the manuscript seeks to address this specific issue and infer the classification status of mask landsat pixels. Its novelty, compared to previous work addressing the same issue, is its unique use of supervised classification to leverage historic inundation frequency (see response to comment #1 from reviewer 2).

**Text modification:**
- Modified manuscript title and abstract to reflect its new scope
- Rewritten introduction to reflect the points made in our response above

4. Relatedly, I also feel this manuscript is lacking some discussion about limitations and specific applications. The discussion about the different assumptions of the method is good; however, I was left wondering more specifically where this method might work and where it might fail. For example, would this method work in areas of high topography/high latitudes where topographic shadowing is an issue? What is the smallest lake this method would work on? Is there a relationship between cloudiness/size/error? I would also advise more discussion about what might have caused the outlier points removed in the time series analysis.

**Response**
We thank the reviewer for their comment and hope that the new result section about the approach's sensitivity to classification errors in the input imagery (Section 3.1) will address their concerns. Many of the questions asked by the reviewer in their comment boil down to the effect of pixel misclassification (or, more fundamentally, to too little or wrong information) on the method's ability to re-classify as land or water the pixels that were previously identified as clouds. The new analyses in the revised manuscript (comments 1-2 above) show that the method is robust to the former (too little information) but can be sensitive to the latter (wrong information).

In practice, this means that the method will not perform well in locations where circumstances (topographic shading, cloud shading, snow/ice, etc) makes it difficult to reliably distinguish water from clouds and land using multispectral imagery. In contrast, the method is likely to be generally robust to application to smaller lake sizes, despite a limited number of unmasked pixels available to train the supervised classification. However, performance is contingent on the assumptions that (i) the location of cloud and water pixels are statistically independent and (ii) that lake bathymetry does not change over time. In the original manuscript we posited that these assumptions are less likely satisfied in small lakes. However, we find no evidence of that in our extended analysis in the revised manuscript, so the hypothesized relationship between lake size and performance is dropped in the revised manuscript.

**Text modification:**

- Modified discussion section (Section 4) focusing on the effect of classification errors on gap-filling performance (L 308-331).

- Modified conclusion section (Section 5) discussing the practical implications of these effects on the applicability of the method (L 367-374)

Specific Comments:
**Response**
We thank the reviewer for their detailed specific comments, which we think generally improve the readability of the manuscript. We will address all comments in the revised manuscript, provided that they still apply (i.e. if the referred text was not already modified to address a major comment).

L1: "The empirical attribution of 'past' rapid hydrologic change" L15: change "when applicable" to "where available"

L15: I would advise adding a sentence at the end of the abstract stating the importance/broader significance of your findings, instead of just stating that your method works

L18: In my opinion, the first few sentences of the paper are weak. I would suggest rewriting slightly (i.e. "Despite their importance, many lakes are undergoing rapid change. . ." makes little sense – the importance of lakes doesn't necessarily mean that lakes will not or should not undergo rapid change). Since "rapidly changing" is a key part of the manuscript, I would also suggest defining what you mean by rapid change paper since the time scale implied by "rapid" can vary based on the reader's background.

L27: This sentence ("By providing") should start the next paragraph, not sit at the end of this one as it interrupts the flow

L31: The paragraph starts by talking about monitoring surface water extent, but then discusses radar altimeters before moving back to extent. I would suggest restructuring this paragraph, or at least the first sentence of it, as the current structure is confusing

L83: I suggest adding a sentence or two to the final paragraph of the introduction stating something like "we test this method over XX lakes, analyze its accuracy and demonstrate its utility" just to provide readers with a better road map for the manuscript

L86: I would call this first step something like "Masking" instead of pre-processing (see major comment #1 above).

Figure 1: I like this figure, but think it could be improved slightly by increasing the size of the image panels and decreasing the size of the arrows and white space. The image panels are hard to see in places and there's plenty of white space so it should be straightforward to make them a bit larger and easier to see.

L149: The sentence starting with "Indeed, visual inspection of satellite. . ." is unclear. I think what the authors are stating is that the area-elevation curves do not match the satellite-observed area, but this section could be clarified.

Figure 3: Please make the x and y labels and the symbols themselves much larger, it is nearly impossible to read the figure at this scale.

L181: "The analysis suggests. . ." this is not a complete sentence, please edit Figure 5: Please make x and y labels larger

L229: change phrase starting with "if shadows. . ." to "shadows covering dry land in the vicinity of the lake may cause an overestimation of the surface area of the lake"

L225-234: Does the cloud masking method remove cloud shadows?

L225-234: Is the influence of topographic shadowing examined? Topographic shadowing, particularly in the NY lake (in winter) could influence classification accuracy (and would not be a randomly distributed error). Even if most of the lakes specifically examined here occur in the tropics or in areas with little-to-no surrounding topography, topographic shadowing issues would likely impact the applicability of this method in other areas and therefore should be discussed

**Referee #2**

In this study, the authors developed a hybrid remote sensing algorithm to estimate the time series of water surface areas over several lakes with rapid changes. The major novelty of this paper is that the proposed algorithm is still workable when the remote sensing images are partially covered by clouds/low-quality pixels. To enhance the capability of working under cloud condition, the authors first utilized unsupervised algorithm to classify pixels with high quality and then create a high-confidence inundation frequency (IF) image. Next, the supervised algorithm and the IF image were used to interpret masked pixels. The validation results against in situ observations indicate that the algorithm is able to monitor water surface changes with a high accuracy. This paper is well-written and easy to follow. However, there are several major concerns related to the following aspects: (1) incomplete literature review (2) the applicability of the algorithm and (3) the design of the validation.

**Response**
We thank the reviewer for the positive comments and hope that our responses to their comments below, and the associated modifications to the manuscript have addressed their three specific concerns.

(1)Some efforts have been made to monitor surface water on global scale using remote sensing. It is not only necessary to acknowledge those work in this paper, but also meaningful to highlight the difference. For example, Pekel et al. (2016) developed a global water map by applying Machine Learning algorithm to Landsat images. Khandelwal et al. (2017) developed a method to monitor global lakes using MODIS images. Very similar to this paper, Zhao and Gao (2018) used an automatic approach to correct the contaminated images for assessment of reservoir surface area dynamics over 6,817 global reservoirs. They used the water occurrence map derived by Pekel et al. (2016) to correct the pixels with low quality, which is very close to the idea of IF image in this paper.

**Response**
We thank the reviewer for their suggestions and have rewritten the introduction to address it (also in response to comment 3 from reviewer 1). In the new introduction we provide an extensive review of the literature (now specifically referring to the mentioned studies, among others), and clarify our contribution as specifically addressing the problem of retrieving classification information from masked pixels. We agree with the reviewer that the approach from Zhao and Gao (Z&G), which we were not aware of, addresses the same problem with a comparable method, although with important differences. Z&G is indeed similar to the proposed approach in that it leverages historical inundation frequency information to infer the flooding status of masked pixels. Both approaches rely on the assumption that all masked pixels with inundation frequency above a certain threshold should be designated as flooded. However, the two approaches differ in the way they define this threshold, which varies from image to image depending on

the current level of the lake. Z&G use a heuristic approach based on the shape of the histogram of unmasked flooded pixels to determine the threshold. In contrast, our approach relies on machine learning (supervised classification) to leverage the relationship between the inundation frequency and current status of *all* unmasked pixels: flooded and unflooded. By relying on statistical relationships embedded within each image, rather than on arbitrary heuristic rules that are not able to vary across images, the proposed approach is more flexible and less reliant on user input. We now explicitly refer to Z&G in the introduction, along with the other approaches that we are aware of that address the same challenge (e.g., Avisse et al 2017 and Khandelwal et al 2017, which both rely on a DEM, and Schwatke 2019), and clarify the specific contribution of the proposed approach.

**Text modification:**
Rewritten introduction to reflect the points made in our response above

2)The applicability of the algorithm needs to be further demonstrated given the fact only a few lakes were tested in this study. Besides this, all the selected lakes are very large considering the 30-m spatial resolution of Landsat. And the majority of the global lakes are much smaller than 1 km2. It would be interesting to add a section discussing the performance of the algorithm over small lakes. We can find the lowest accuracy was obtained over the smallest lake in this study which raises a question - if the accuracy goes down when the size of lakes decreases.

**Response**
We agree with the reviewer that the (apparent) relationship between lake size and prediction accuracy in the original manuscript merited further analysis. In the original manuscript, we posited that the independence and stationarity requirements of the approach are less likely to hold for lakes. However, this hypothesis was dropped in the revised manuscript because additional analyses (both empirical and numerical) provide no evidence to support it:

**(i) Empirical Analysis:** We added two validation lakes in the revised manuscript, which size (~1km2) is much smaller than the previously considered lakes. The gap filling algorithm performed well on both added lakes, which are located in a flat and semi-arid region in Texas. This suggests that the poorer performance on the smallest lake in the original manuscript (Cannonsville reservoir, NY) was due to local topography and climate,  rather than its size.  (Unfortunately, while most global lakes are indeed smaller than 1km2, very few have openly available in-situ data of their surface area at the frequency and coverage period considered in this study.)

**(ii) Numerical Experiment:** We also conduct a numerical experiment in which we explore multiple ways in which the accuracy of our approach will be affected, representing a variety of classification and practical challenges. The method was quite sensitive to classification errors in the input imagery (e.g., clouds misclassified as water, water misclassified as land, etc), which we interpret as indicative of its limitation in situations where circumstances (topography, climate) makes it difficult to systematically identify water on the input multispectral imagery. These effects might have confounded the effect of lake size in the original manuscript, as previously argued (empirical analysis). Conversely, we find that the method's ability to predict the class (water or land) of the artificially masked images is robust to an increasingly large fraction of masked images -- within certain limits described below.

Small lakes, in particular, will be affected both by the limited amount of information to conduct the training for gap-filling as well as the fact that edge pixels will be mixed water and land, and as lakes decrease in size the relative proportion of edge pixels increases. Both of these issues are addressed in our synthetic analysis of the 1 km2 lake in which we increased the proportion of masked pixels (up to 90%) and consider the effects of inaccurate classification (this latter effect can be considered a proxy for the increasing proportion of lake edge pixels and misclassification around the edge). When the area of unmasked pixels is sufficiently large (> 25 ha), the gap-filling accuracy is generally robust to variations in cloud cover or lake size. However, when the the area of unmasked pixels drops below 25 ha, gap-filling accuracy becomes highly sensitive to classification accuracy and the number of unmasked pixels.

Together, these results suggest that water detection is the primary limitation for large lakes, and that lake size (and fraction of the lake that is masked) stands as an additional limitation when the unmasked area becomes sufficiently small (roughly below 25 ha in our synthetic analysis)..

**Text modification:**

- Added validation lakes in Figure 4 and S1
- Added sections describing the numerical experiments (Section 2.2) and their results (Section 3.1)
- Added discussion section on the effect of classification errors on gap-filling performance (L308-331). The respective effects of  "too little" information due to cloudiness, an overly eager cloud detection method or a smaller lake size, or of "wrong information due to water-land classification errors are specifically discussed, as well as the interaction between the two for small lakes.

(3)To validate the results, the authors compared all remotely sensed water extents with in situ observations. However, one of the highlights of the proposed approach is to interpret the masked pixels. So I am wondering how the algorithm performs during the cloudy season when lots of pixels are masked.

**Response**
We thank the reviewer for their suggestion, and have added to the revised manuscript a figure for each validation lake representing lake extent estimation error against  cloudiness. We find that errors are not significantly correlated to the extent of cloud cover if clouds are appropriately detected., This finding is in line with our interpretation that the approach is robust to having too little (small or heavily clouded lakes), information in the input imagery. However, our numerical experiment also indicates that errors are correlated to the prevalence of *wrong* information (misclassification of water or land) in the input imagery. This indicates that overly greedy cloud/shadow masking algorithms will yield better results compared with algorithms that underpredict cloud/shadow pixels. This also entails that challenges will arise when many cloud pixels are difficult to detect (e.g., waspy clouds or cloud edges).

**Text modification**
- Added scatterplots of errors vs. cloudiness in Figures 4, 5 and S2.
- Added discussion section on the effect of "too little" information due to cloudiness (L308-331).

Other specific comments:
Thank you for your comments, which we will be sure to address in the revised manuscript.

(1) Line24, miss a space in 'consequenceof'.
(2) It would be better to increase the font size for all figures.

---

## Author Response (AR2)

Dear Dr Kelleher,

Please find attached the revised version of our previously submitted major revision of the manuscript: "A simple cloud-filling approach for remote sensing water cover assessments", for consideration as a research article at HESS.

We have carefully addressed the concerns voiced by yourself and the reviewer and are grateful for the opportunity to resubmit. Thank you for taking the time to review the major revision and for the comments you both provided. A point-by-point response (in blue) to all reviewers' comments and the related revised text (in red) are provided below. A track-change version of the manuscript and SI is enclosed.

Thanks again for your consideration. Sincerely,

Marc Muller, on behalf of all authors.